

# California air resources board forest carbon protocol invalidates offsets

Bruno D.V. Marino[1], Martina Mincheva[2] and Aaron Doucett[3]

[1] Executive Management, Planetary Emissions Management Inc., Cambridge, MA, USA
[2] Department of Statistics, Philadelphia, PA, United States of America
[3] Planetary Emissions Management Inc., Cambridge, MA, USA

## ABSTRACT

The commercial asset value of sequestered forest carbon is based on protocols employed globally; however, their scientific basis has not been validated. We review and analyze commercial forest carbon protocols, claimed to have reduced net greenhouse gas emissions, issued by the California Air Resources Board and validated by the Climate Action Reserve (CARB-CAR). CARB-CAR forest carbon offsets, based on forest mensuration and model simulation, are compared to a global database of directly measured forest carbon sequestration, or net ecosystem exchange (NEE) of forest $CO_2$. NEE is a meteorologically based method integrating $CO_2$ fluxes between the atmosphere, forest and soils and is independent of the CARB-CAR methodology. Annual carbon accounting results for CAR681 are compared with NEE for the Ameriflux site, Howland Forest Maine, USA, (Ho-1), the only site where both methods were applied contemporaneously, invalidating CARB-CAR protocol offsets. We then test the null hypothesis that CARB-CAR project population data fall within global NEE population values for natural and managed forests measured in the field; net annual $gC\ m^{-2}yr^{-1}$ are compared for both protocols. Irrespective of geography, biome and project type, the CARB-CAR population mean is significantly different from the NEE population mean at the 95% confidence interval, rejecting the null hypothesis. The CARB-CAR population exhibits standard deviation $\sim 5\times$ that of known interannual NEE ranges, is overcrediting biased, incapable of detecting forest transition to net positive $CO_2$ emissions, and exceeds the 5% CARB compliance limit for invalidation. Exclusion of $CO_2$ efflux via soil and ecosystem respiration precludes a valid net carbon accounting result for CARB-CAR and related protocols, consistent with our findings. Protocol invalidation risk extends to vendors and policy platforms such as the United Nations Program on Reducing Emissions from Deforestation and Forest Degradation (REDD+) and the Paris Agreement. We suggest that CARB-CAR and related protocols include NEE methodology for commercial forest carbon offsets to standardize methods, ensure *in situ* molecular specificity, verify claims of carbon emission reduction and harmonize carbon protocols for voluntary and compliance markets worldwide.

Corresponding author
Bruno D.V. Marino,
bruno.marino@pem-carbon.com

![PeerJ]

## INTRODUCTION

We review and analyze California Air Resources Board (CARB) protocols representing net annual forest carbon sequestration (*Marland et al., 2017*) to characterize carbon accounting uncertainty, and ultimately how well the protocol reflects actual net forest carbon sequestration compared to independent and direct measurement of carbon. The CARB protocols must quantify net carbon sequestration as real, additional, permanent, verifiable, and enforceable (*California Air Resources Board, 2011*; *California Air Resources Board, 2014*; *California Air Resources Board, 2015b*), a requirement to ensure that erroneous offsets (e.g., Air Resources Board Offset Credits, or, ARBOC's) do not compromise the integrity of California's AB32, and its successor AB398, mandatory cap on emissions (*Nunez, 2016*; *Schatzki, Stavins & Hall, 2018*). The CARB issues carbon offsets based on determination of net forest carbon sequestration according to CARB and Climate Action Reserve (CAR) protocols (CARB-CAR); ~123 million metric tons of $CO_2$ equivalent ($CO_2$e)[1] have been issued since 2006 (*California Air Resources Board, 2018a*). Forest offsets account for up to ~79% of CARB offsets issued, emphasizing the importance of their validation (*California Air Resources Board, 2018a*). The CARB-CAR compliance rules stipulate invalidation criteria for offsets that exceed 5% of actual net forest carbon (e.g., overcrediting) for a period of up to eight years (*California Air Resources Board, 2015b*; *California Air Resources Board, 2015a*) and correction of errors due to material misstatement (e.g., percentage error; *California Air Resources Board, 2013b*)[2]; these criteria have not been widely applied to CARB-CAR offsets. The CARB-CAR protocols and resulting registered offsets have not been independently validated approximately thirteen years since California passed the Global Warming Solutions Act (AB32) in 2006 with the goal of reducing its greenhouse gas (GHG) emissions to 1990 levels by 2020 (*Nunez, 2016*). Moreover, the lesson learned from false claims of emission reduction by the automotive industry is that all commercial claims of emission reduction must be verified by independent direct measurement (*Li et al., 2018a*) to ensure product value and integrity.

We address the validation gap by comparing CARB-CAR estimated results (e.g., $CO_2$ is not observed directly) with a global database of *in situ* measured net forest carbon sequestration flux, or net ecosystem exchange (NEE) (*Baldocchi, Chu & Reichstein, 2018*; *Baldocchi & Penuelas, 2019*; *FLUXNET, 2019*). Three assumptions of the CARB-CAR protocol are evaluated in this review: (1) offsets represent complete forest carbon accounting, (2) results are compliant with CARB-CAR 5% invalidation rule for overcrediting, and (3) results reflect actual net annual forest carbon ecosystem dynamics resulting in verified forest carbon financial products. We present data for selected projects, such as the Howland Forest, Maine, USA, the only location where both methods were applied, and for statistical tests to compare CARB-CAR populations with directly measured forest carbon sequestration population data obtained in the field by the NEE method. The results characterize the uncertainty, accuracy and precision (e.g., interannual variance) of the CARB-CAR protocols, their effect on pricing of forest carbon products and their impacts on carbon market integrity. Protocols related to the CARB-CAR methodology are also assessed for uncertainty and impact on forest carbon markets.

[1] The concentration of $CO_2$ that would cause the same amount of radiative forcing as a given mixture of $CO_2$ and other greenhouse gases.

[2] *percentage error* $= \frac{|Estimated\ value - NEE1\ value|}{NEE1 value} \times 100$ Where estimated value = CARB-CAR, and NEE1 value = directly measured value. We propose recalculation of material misstatement errors based on NEE values as follows: Percent error determination for offsets based on the material misstatement equation (*California Air Resources Board, 2013b*) is defined as follows: %*error* $= \sum \left[ \frac{Discrepancies + Omissions + Misreporting}{Total\ reported\ emissions} \right] \times 100\%$. Simplifying, where *Discrepancies* are defined as the CARB-CAR value, *Total reported emissions* are defined as NEE1 reported emissions, NEE1 represents the accepted value, and *Omissions and Misreporting = 0,* the expression simplifies as,

*percentaage error*

$= \frac{|CARB_{CAR} - NEE\ value|}{NEE1\ value} x100$
[3]**CARB-CAR**. The 63 project types include avoided conversion (AC, $n = 7$), reforestation (RF, $n = 2$), and improved forest management (IFM, $n = 54$) projects (Table 1). AC and RF projects involve temperate zone historically forested areas and must maintain forest stocks through CARB-CAR regulatory practices (*California Air Resources Board, 2011*; *California Air Resources Board, 2014*; *California Air Resources Board, 2015b*). Reforestation projects involve land that has been deforested (e.g., canopy cover less than 105) and is actively replanted with trees. Details of management practices for each site vary, however, all sites comply with prohibition of clear-cut (e.g., <40 acres harvested), broadcast fertilization, extensive soil disturbance and extensive short harvest rotations typical of timber plantations, ensuring comparability of baseline management practices and application of the compliance protocol process as regulated by the CARB-CAR protocols (*California Air Resources Board, 2011*; *California Air Resources Board, 2014*; *California Air Resources Board, 2015b*). Aggressive management practices involving timber removal would trend towards more positive CARB-CAR values (e.g., net positive emissions to the atmosphere) rendering the CARB-CAR data base conservative for comparison with NEE values. The IFM and AC projects ($n = 61$) are subject to similar land management activities, including minimal harvesting, as defined by the protocols. CARB-CAR net forest carbon sequestration for the 63 projects is characterized by a mean and standard deviation of $-948.8 \pm 1,504.8$ ('Box Plots'). Descriptive reports and cumulative emissions data are derived from the CAR website homepage (*Climate Action Reserve, 2018a*). Forest carbon results for the American Carbon Registry (*American Carbon Registry, 2018*) and the Verified Carbon Standard (*Verified Carbon Standard, 2018*), both approved project registries by the ARB, were reviewed but were not included in this study. ACR project data was not available in summary format for carbon offsets presenting a challenge to verify ACR results. A total of 55 ACR forest carbon projects were listed as of 09-01-2018; 18 identify values for registered carbon credits but serial numbers for ARB issued offsets were not available (*American Carbon Registry, 2018*). Analysis of the ACR Part VII forest project listing applications, identical to those for CARB-CAR applications, *(Note continued on next page.)*

# LITERATURE SURVEY, DATA SOURCES AND METHODS

## CARB-CAR

The CARB-CAR population data used in this study represent 63 sites covering 340 site years, primarily located in the US temperate zone, and have been assigned CAR serial numbers to project offsets (units: $gC\ m^{-2}yr^{-}$) (Table 1, Supplemental Informations 1,3–7 for site information and data). The CARB-CAR projects listed in Table 1 were sourced from the California Air Resources Board and the California Environmental Protection Agency website pages as noted for "Early Action Projects" (*California Air Resources Board, 2015c*) and as ARBOC's issued as of 09-01-2018, and must adhere to CARB-CAR forest management protocols. Forest project data were accessed through links to a Climate Action Reserve project identification number (CAR#) providing a project summary page, a document summary page and a cumulative performance report page. The data used in this analysis was sourced directly from the Cumulative Performance Report page and from the column of Verified GHG Reductions for each year of each project or as otherwise reported on the project page when a Cumulative Performance Report was not available (Table 1). We reviewed CARB-CAR registry records and history for each project noting discretionary inconsistencies in protocol reporting and process (Table 1). The CARB protocols and accounting requirements are described in three related primary documents published in 2011 (*California Air Resources Board, 2014*), 2014 (*California Air Resources Board, 2014*) and 2015 (*California Air Resources Board, 2015b*) entitled: "Compliance Offset Protocol U.S. Forest Projects". The underlying equations for the CARB-CAR protocol are reviewed (Supplemental Information 8) to identify carbon accounting terms and to establish similarities with related protocols including the American Carbon Registry (ACR) (*Winrock International, 2016*), the Clean Development Mechanism (CDM) (*Warnecke, 2014*) and the Verified Carbon Standard (VCS) (*Verified Carbon Standard, 2018*). The CARB-CAR protocols employ limited forest mensuration practice (e.g., forest survey every six years or longer, up to 12 years; *California Air Resources Board, 2015b*; *Marland et al., 2017*), vegetation proxies (*Cawrse et al., 2009*), estimated baselines and growth simulation models (*California Air Resources Board, 2011*; *Climate Action Reserve, 2018b*) to determine net forest carbon sequestration. Direct measurement of $CO_2$ in the field does not occur at any point of the protocol process. Additional details are available in the endnotes.[3]

## NEE

NEE population data, described by *Baldocchi, Chu & Reichstein (2018)*, and *Baldocchi & Penuelas (2019)*, herein referred to as NEE1 and NEE2 (units: $gC\ m^{-2}yr^{-1}$), respectively, were employed to assess validity of CARB-CAR results representing direct measurement of $CO_2$ flux. The NEE1 population data represent 59 eddy covariance tower sites, primarily within the US and Canada, with a total of 540 site years. The NEE1 project annual data, employed for the analyses presented, was sourced from each of the NEE1 site references and cross-checked with the NEE1 published data for each site. (see Supplemental Information 1, 3–7 for site information and data) and utilized for comparison with CARB-CAR project interannual data. NEE is also expressed as Net Primary Productivity (NPP) where the term—NEE is employed (*Chapin et al., 2006*). NEE is a meteorologically based direct
*(Note 3 continued from previous page.)* verified that soil carbon was not included in the carbon pools employed in net forest carbon estimations (*n* = 30: 189, 173, 192, 199, 202, 211, 298, 249, 200, 256, 265, 266, 267, 268, 262, 265,255, 276, 282, 282, 273, 274, 284, 288, 292, 303, 324, 360, 277, 278.). The VCS identified 12 proposed forest projects, however, offset credit summaries and serial numbers for ARB issued offsets were not available. Based on the information available, ACR and VCS results were not considered in this study. CARB-CAR defines forests in terms of photosynthesis and carbon stored in soils and litter. A forest project is a planned set of activities designed to increase removals of $CO_2$ from the atmosphere, or reduce or prevent emissions of $CO_2$ to the atmosphere, through increasing and/or conserving forest carbon stocks. A net climate benefit, or verification of a net negative $CO_2$ balance for the forest ecosystem must result from the project activities. The Offset Project Boundary defines GHG emission sources, GHG sinks, and GHG reservoirs that must be accounted for and quantified for a Forest Project's GHG reductions and removal enhancements. Tables in the CARB-CAR protocols provide a comprehensive list of project components that must be included in the Offset Project Boundary for each type of Forest Project. A "reservoir/pool" is accounted for by quantifying changes in carbon stock levels. GHG sources or GHG sinks, GHG reductions and GHG removal enhancements are accounted for by quantifying changes in GHG emission or GHG removal enhancement rates. The CARB protocols and accounting requirements are described in three related primary documents published in 2011 (*California Air Resources Board, 2014*), 2014 (*California Air Resources Board, 2014*) and 2015 (*California Air Resources Board, 2015b*) entitled: "Compliance Offset Protocol U.S. Forest Projects". The cited protocols identify field methods employing forest mensuration surveys, model simulation requirements and primary sources, sinks and reservoirs in a series of tables for Reforestation Projects (RF), Improved Forest Management Projects (IFM) and Avoided Conversion Projects (AC). The CARB and CAR forest carbon protocols, while published in separate documents (*California Air Resources Board, 2015b*; *Climate Action Reserve, 2018b*) rely on shared estimation protocols for net GHG reductions and removals. For example, *(Note continued on next page.)*

measurement method, deployed in the field, integrating all forest $CO_2$ fluxes (e.g., above and below ground) and is methodologically independent of the model and estimation-based CARB-CAR protocol. We provide a graphical interpretation of the NEE1 data in Supplemental Information 9, illustrating the relationship and relative magnitude for NEE (open rectangle, right axis), GPP and $R_{eco}$ (open circles), filled and gray rectangle symbols represent NEE1 data for the Howland Forest, Maine, USA (Ho-1), and the Wind River (Wrc), Washington, USA, sites, respectively, illustrating overlapping and narrow ranges for diverse forests (e.g., west coast redwoods versus east coast hardwoods). NEE, GPP and $R_{eco}$ values falling outside of the known ranges likely do not reflect natural and managed forest ecosystems and are a sign of inaccurate estimates and a basis for invalidation. NEE2 data, inclusive of NEE1 data, expand the GPP, $R_{eco}$ and NEE population database for ecologically diverse natural and managed forests across 155 global observation platforms representing 1,163 site years. NEE1,2 data are derived from measurement of gaseous molecular $CO_2$ vertical fluxes (*Baldocchi, Chu & Reichstein, 2018*; *Baldocchi & Penuelas, 2019*) and are based on globally applied and standardized eddy covariance methods (*Baldocchi, 2003*; *Baldocchi, 2014*) to quantify NEE (*Chapin et al., 2006*; *Luyssaert et al., 2009*) as $tCO_2e$ or gC m$^{-2}$yr$^{-1}$. NEE data, available from the Fluxnet database (*Fluxnet, 2019*), are routinely employed to test significance of trends for forest growth (*Ney et al., 2019*), soil and ecosystem respiration (*Bond-Lamberty & Thomson, 2010*; *Göckede et al., 2019*), fire disturbance (*Rocha & Shaver, 2011*) and effects of climate change on diverse ecosystems (*Yu et al., 2019*), similar to the analytical approach employed in this report. Additional information for NEE projects is described in the endnotes.[4]

## Statistical calculations

Individual annual records (units: gC m$^{-2}$yr$^{-1}$) were used in this analysis; trends in time series are not considered. Our analysis is based on population data for NEE, GPP and $R_{eco}$ (NEE1,2) representing primarily diverse temperate zone forest ecosystems (Supplemental Information 9) of the world. The tight coupling between GPP and $R_{eco}$ constrains NEE within relatively narrow boundaries (mean, standard deviation) considering the global spatial and annual time scales reported (*Baldocchi & Penuelas, 2019*); values falling outside the NEE1,2 boundaries are likely inaccurate, and with additional criteria may be considered invalid. Although only a single location, the Howland Forest, shares application of both NEE and CARB-CAR protocol, an accurate measurement of net carbon sequestration for any location is expected to comply with the NEE, GPP and $R_{eco}$ relationship (Supplemental Information 9, filled and gray rectangle symbols for Howland Forest and Washington, GPP and $R_{eco}$, respectively). Each data point (whether in NEE or CARB-CAR) comes from the underlying population of the world's forests (e.g., managed or conserved) providing the basis for the two-sample statistical tests presented. We employ standard statistical tests to compare population characteristics, including the accuracy in means and precision (i.e., interannual range) for the CARB-CAR protocol methods, relative to directly measured NEE1,2 sample annual values and variance. See Supplemental Information 2 for details of the statistical analyses.
*(Note 3 continued from previous page.)* both protocols employ forest growth simulation models that use the Forest Vegetation Simulator (FVS) and related vegetation proxies for forest project species. FVS features are coupled with identical numerical equations and carbon pool terms for net forest carbon sequestration directly linking the CARB and CAR protocols (e.g., CARB-CAR) (*California Air Resources Board, 2011*; *California Air Resources Board, 2014*; *California Air Resources Board, 2015b*; *Climate Action Reserve, 2018b*) (Supplemental Information 8). The underlying models and their specific application to the CARB project location are detailed in documents associated with each of the CARB projects as listed on each CAR project page. Table 1 provides features of the CARB-CAR data sets that appear to be discretionary and are applied inconsistently across the CARB-CAR project sites. CARB-CAR cites seven approved forest growth and yield models (*California Air Resources Board, 2019*); otherwise, shared standards and references are lacking. Details of the ARB Compliance Offset Program and offset credits issued are provided by the CARB website (*California Air Resources Board, 2018a*). All approved verification protocols must adhere to CARB standards (*California Air Resources Board, 2013a*). The CARB-CAR protocol specifies the soil carbon reservoir/pool as item 6 listed as RF-6, IFM-6 and AC-6. The soil carbon information applicable to the protocols are listed in Tables 5.1 (RF-6), 5.2 (IFM-6) and 5.3 (AC-6) for the protocols published in 2011 (*California Air Resources Board, 2014*) and 2014 (*California Air Resources Board, 2014*), respectively. The same information is listed in Tables 4.1 (RF-6), 4.2 (IFM-6) and 4.3 (AC-6) in 2015 (*California Air Resources Board, 2015b*). Reference to inclusion or exclusion of the soil reservoir for each project listed in Table 1 is indicated and linked to one of the above protocols as cited in the summary documents provided for each project. Forest mensuration, or biometric methodology, is intrinsic to the CARB-CAR protocol process and outcomes and is briefly reviewed here. Timber surveys, designed for timber operations, are required every six years or longer (*California Air Resources Board, 2014*; *California Air Resources Board, 2015b*), however, simulation models estimate annual *(Note continued on next page.)*

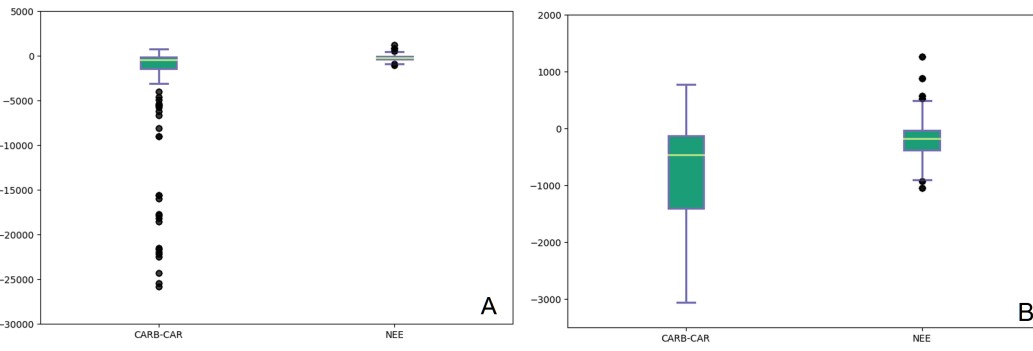

**Figure 1  Box plots of annual records from CARB-CAR (340 years, 63 sites) and NEE1 (540 years, 59 sites) projects.** (A) The box plots show the median (white line through each box), the 25th percentile (bottom of lower box), the 75th percentile (top of upper box), the upper and lower whiskers represent the upper and lower values that are not outliers, and outliers (individual closed circles). The CARB-CAR data show outliers exceeding $-12,000$ gC m$^{-2}$yr$^{-1}$. CARB-CAR median is $-445.4$ gC m$^{-2}$ yr$^{-1}$ compared to the NEE1 median value of $-172.5$ gC m$^{-2}$ yr$^{-1}$. The means and standard deviations ($\pm$) are, $-948.8 \pm 1,504.8$ and $-198.2 \pm 261.6$, for CARB-CAR and NEE1, respectively. (B) Box plots, described as above for CARB-CAR and NEE1 populations with CARB-CAR outliers removed.

## RESULTS

### Box plots

Figure 1A shows the difference in distribution, measures of central tendency and outliers between the CARB-CAR and NEE1 populations for net forest carbon sequestration. A selected interval of the box plot excluding the CARB-CAR outliers is presented in Fig. 1B to illustrate the population differences (CARB-CAR median of $-445.4$ gC m$^{-2}$yr$^{-1}$ compared to the NEE1 median value of $-172.5$ gC m$^{-2}$yr$^{-1}$ ) corresponding to the larger spread and left-skewness of CARB-CAR values. The sample means and standard deviations ($\pm$), for all annual values, Fig. 1A, of the CARB-CAR and NEE1 datasets are, respectively, $-948.8 \pm 1,504.8$ gC m$^{-2}$yr$^{-1}$ and $-198.0 \pm 261.6$ gC m$^{-2}$yr$^{-1}$ representing an extreme range of $\sim 5\times$ the value for CARB-CAR forest carbon sequestration mean and $\sim 6\times$ standard deviation for interannual variance relative to the NEE1 population data (*Baldocchi, Chu & Reichstein, 2018*). The difference in mean values between the two populations is significant at the 95% confidence level, rejecting the null hypothesis that the CARB-CAR population mean falls within the NEE1 population mean. The mean and standard deviation of the NEE2 data are $-156 \pm 284$ gC m$^{-2}$yr$^{-1}$, respectively (155 sites; 1,163 site year), supporting the comparison with CARB-CAR data and $\sim 6\times$ the CARB-CAR mean and $\sim 5\times$ the standard deviation characterizing interannual ranges for natural and managed forest ecosystems. NEE1,2 natural variation for global forests across management activities is constrained tightly by GPP and R$_{eco}$ (Supplemental Information 9) providing boundaries for NEE comparisons.

### Difference between means

Figure 2 shows a plot of the 95% confidence interval for the difference in means between the CARB-CAR and NEE1 population of (*Baldocchi, Chu & Reichstein, 2018*) annual

*(Note 3 continued from previous page.)* incremental change for CARB-CAR net forest carbon sequestration absent annual surveys. CARB-CAR Forest mensuration methods rely primarily on measurement of tree diameter at breast height (DBH) (*Gonçalves et al., 2017*). The limitation of the biometric approach is that biomass is not directly measured as it is not quantified by harvest and weight of the carbon pools; this approach is not practical or economically feasible resulting in destruction of the forest. Uncertainties of 50%–80% for individual trees and 20+ % for plot level estimation persist for forest mensuration (*Gonçalves et al., 2017*; *Holdaway et al., 2014*; *Paré et al., 2015*). Timber survey errors include: (1) variation in the parameters of allometric equation(s) and natural variability of tree structures, (2) measurement errors (DBH, tree height) and differences in frequency of measurement (e.g., multiple measurements per year), and (3) selection of tree-specific parameters within allometric equations such as wood density. The uncertainties are compounded when the forest areas have been or are subject to management including timber extraction, thinning and prescribed or natural fire. In many cases diverse sources of uncertainty are not identified, or new sources of uncertainty are introduced due to bias in data collection, limited coverage of representative forest areas, exclusion of selected carbon pools and inconsistent application of standards and calibration of equipment between measurements. An example of a comprehensive forest mensuration protocol is found in *Barford et al. (2001)* (e.g., weekly measurement of DBH during the growing season, biomass calculation using density data from a study of northern hardwood forests similar in latitude and elevation, weekly collection of leaf litter during the fall months sorted by genus, dried and weighed. $CO_2$ ecosystem and soil respiration cannot be estimated from forest mensuration methods. Examples of forest mensuration, including soil $CO_2$ and ecosystem respiration efflux, in relation to eddy covariance approaches are well represented (*Curtis et al., 2002*; *Luyssaert et al., 2009*; *Ouimette et al., 2018*).

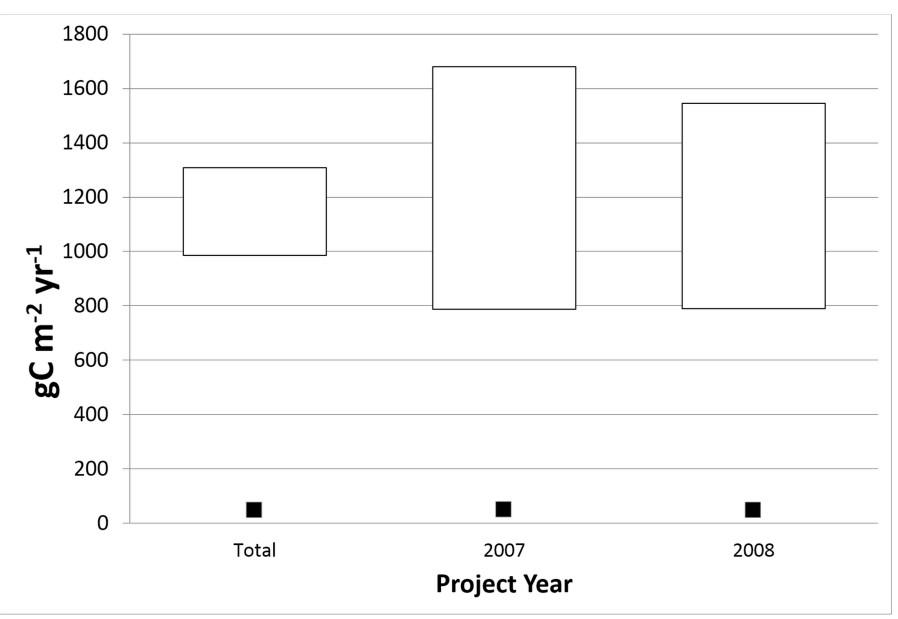

**Figure 2** **Plot of 95% confidence interval for the difference in means between CARB-CAR and NEE1 annual data.** The combined data set (All Years) consists of 340 CARB-CAR and 540 NEE1 data points. A formula for a large-sample confidence interval (described in Methods) is used for the unfilled bars and no assumption on equal standard deviations between the two data sets has been made. The top and bottom of the open bars represent the range of the difference between the CARB-CAR and the NEE1 means with a 95% confidence level. The filled square symbol below each bar represents the 5% estimation error allowed by CARB-CAR.

measurements for all years available and for the selected years of 2007 and 2008. Given the large difference in sample means for the CARB-CAR and NEE1 datasets the true population means may also be significantly different. We test the null hypothesis that the two sample populations were drawn from the same underlying population of annual values for forests. The combined data set consists of 340 CARB-CAR and 540 NEE1 data points ("Total"), each reported as representing an annual cycle determined by each protocol. A formula for a large-sample confidence interval is used for the bar labeled "Total"; no assumption on equal standard deviations between the two data sets has been made. Amongst the years with overlapping data (2001–2014), we choose 2007 and 2008, as they have the largest number of combined sample points, 65 in 2007 (23 for ARB and 42 for NEE1) and 65 in 2008 (24 for ARB and 41 for NEE1), excluding initial year values. The null hypothesis that the two data sets come from the same population is rejected implying that CARB-CAR protocols reflect anomalous values and cannot be relied upon to manage forest carbon sequestration. The 5% estimation error does not overlap with the 95% confidence interval demonstrating that the CARB-CAR estimates are more than the allowed 5% from the NEE1 population values. The standard deviation for the CARB-CAR data is very large compared to the NEE1 standard deviation, irrespective of the year. For example, in 2008, the standard deviations for CARB-CAR and NEE1 were respectively, 1,170 and 255 gC $m^{-2}yr^{-1}$, a ~5× over-estimation difference for interannual NEE1 values. This leads to
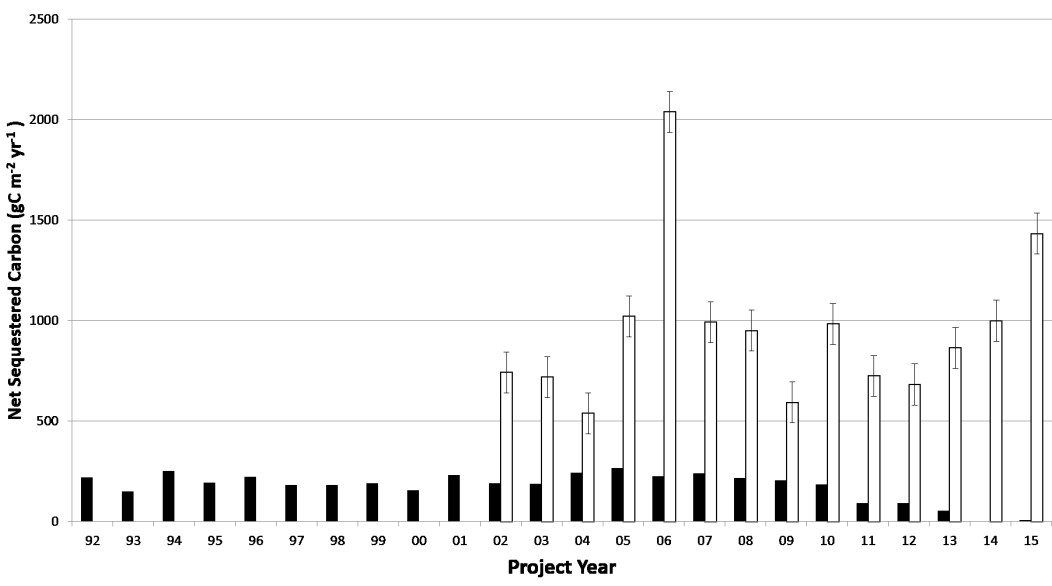

**Figure 3  Time interval plot of CARB-CAR data (open bars) from 2002 to 2015 and NEE1 measurements (filled bars) from 1992 to 2015.**  Values are plotted as positive numbers representing net sequestration of carbon. The averages for the two data sets are shown by each bar representing forest carbon sequestration calculated annually over all available locations. The error bars represent 5% of the CARB-CAR year for 2006 and applied to all CARB-CAR project annual averages.

[4]**NEE**. The NEE1 study reported a mean and standard deviation of approximately $-200.00 \pm 162$ gCm$^{-2}$ yr$^{-1}$(*Baldocchi, Chu & Reichstein, 2018*), compared to a mean of $-198.0$ and standard deviation of $261.6$ gCm$^{-2}$ yr$^{-1}$ calculated in this study. The difference in mean and standard deviation result from different approaches to calculation. We calculated the sample mean and sample standard deviation of the pooled annual data of 544 points (Supplemental Information 7). The authors of the NEE1 data calculate the mean by first obtaining the 59 means corresponding to the 59 different locations, and then calculating the mean of means (*Baldocchi, Chu & Reichstein, 2018*). The NEE1 standard deviation is also based on the deviations of each data point from the corresponding location mean (as opposed to the global mean). The difference in mean and standard deviation noted do not change the conclusions of the summary study (*Baldocchi, Chu & Reichstein, 2018*) or our use of that data. Additional data for the Howland Forest (CARB-CAR671, NEE1-Ho1) covering the years 2008 to 2013 that were not included in the NEE1 dataset were obtained from an additional publication (*Hollinger et al., 2016*); these values were not included in summary statistics reported. We note that although 544 site years was reported, four of those site years were absent data; we use the available 540 site year values and identify the four years of missing data in Supplemental Information 7. Data Availability Statement: NEE annual Data cannot be shared publicly because of Fluxnet2015 (https://fluxnet.fluxdata.org/data/data-policy/) and Lathuile (https://fluxnet.fluxdata.org/data/la-thuile-dataset/) data policies. However, one can access these data in the same manner as the authors via the Fluxnet2015 dataset (https://fluxnet.fluxdata.org/ data/fluxnet2015-dataset/) and the LaThuile dataset (https://fluxnet.fluxdata.org/data/la-thuile-dataset/). The NEE method is based on direct and fast measurements (e.g., 10 Hz) of actual gas transport characterized by a three-dimensional wind field in real time. In this study, we did not control for tower height or upscaling results across the diverse site locations, however, all eddy covariance platforms were positioned above the canopy to capture $CO_2$ flux dynamics. The concentration of the gas of interest (e.g., $CO_2$, $^{13}CO_2$ and $^{14}CO_2$) is measured concomitantly resulting in flux of the gas. Flux data are first  (*Note continued on next page.*)

a very wide confidence interval that also establishes that the CARB-CAR project data are invalid based on the permitted 5% compliance margin of error. To our knowledge, CARB-CAR compliance testing of project results has not been reported employing directly measured forest $CO_2$.

## Project annual interval plot

Figure 3 shows a time interval plot of CARB-CAR annual data from 2002 to 2015 and NEE1 (note: NEE values are illustrated as positive for the purposes of this graph) annual measurements (*Baldocchi, Chu & Reichstein, 2018*) from 1992 to 2015 to further test the invalidation of CARB-CAR offsets according to the 5% invalidation compliance rule. The averages for the two data sets are shown by each vertical bar representing forest carbon sequestration calculated annually over all available locations. The selected intervals are absent first year data for the CARB-CAR population to present a conservative case for testing the null hypothesis. The year 2006 was selected, in which the largest average carbon offset by CARB-CAR sites ($n = 12$) has been recorded, namely $-2,038$ gC m$^{-2}$yr$^{-1}$, and apply the corresponding 5% admissible error of 101.9 gC m$^{-2}$yr$^{-1}$ to all CARB-CAR years, shown as error bars for each CARB-CAR year. No intersection between the admissible interval and the actual NEE1 measurements for any of the overlapping years (2001–2015) is observed. The NEE1 averages through the interval are consistent relative to the large year-to-year fluctuations observed for the CARB-CAR dataset. The null hypothesis is rejected; the CARB-CAR data are invalid by exceeding the 5% compliance threshold for the years represented in this analysis. The absence of negative values, in this case
(*Note 6  continued from previous page.*) converted to half-hourly mean grams of carbon per square meter ($gC\ m^{-2}s^{-1}$) and then summed for each year as the cumulative annual net carbon exchange ($gC\ m^{-2}yr^{-1}$). Tower based estimates of net ecosystem exchange are reported in negative units reflecting a micrometeorological sign convention where flux from the atmosphere is negative, unless otherwise noted. The EC method has been applied worldwide under remote and harsh conditions employing solar power often for months without maintenance (*Burba, 2013*). Open or closed path gas analyzers (e.g., $CO_2$, $CH_4$, $N_2O$) using shared gas standards coupled with automated flux calculation, telemetry and integrated micrometeorological sensors, for example, are typical and deployed across numerous field platforms readily delivered to the project site. EC data are analyzed by a variety of models across small and large scales to calculate NEE (*Burba, 2013*). The spatial footprint of the NEE observation scales with height of sampling inlet above the canopy, representing from ∼1 $km^2$ to ∼10 $km^2$ for typical single EC platforms. Upscaling of EC data provides up to 100 $km^2$ of carbon sequestration data coverage (*Kumar et al., 2016*; *Palmer et al., 2018*; *Richardson et al., 2012*; *Stanley et al., 2018*). Annual errors in NEE typically range between 30 and 100 $gC\ m^{-2}\ yr^{-1}$(*Baldocchi, 2008*; *Burba, Madsen & Feese, 2013*; *Hagen et al., 2006*). Gas analyzers for EC measurements are available from a variety of vendors (e.g., Los Gatos Research, San Jose, CA, USA). The CARB-CAR and NEE project locations are diverse across similar biomes, overlapping in one case of the Howland Forest, Maine, USA, (See Supplemental Information 10 for Howland Forest site information).

representing $CO_2$ emissions to the atmosphere, implies that the CARB-CAR protocol is biased against detection of a switch from net sequestration to net positive emissions, arguably a criterion for invalidation.

### *p*-values table

Table 2 shows the results of a null hypothesis that the difference between the CARB-CAR and the NEE1 annual means is under the allowed 5% threshold. A year-by-year detailed comparison between the population data sets to further test the 5% invalidation threshold for the CARB-CAR data is presented. The test is performed separately for all years between 2002 and 2015; *p*-values are recorded in the last two rows of Table 2. The *p*-values range from 0.00 to 0.065. Typically, *p*-values ≤5% demonstrate a rejection of the null hypothesis. The results reject that the estimation error is within the allowed 5% value, with three exceptions to the 14-year record. In the case of years 2004, 2013 and 2014, the *p*-values are higher than 5% (i.e., 6.53%, 5.48% and 5.24%). However, in these cases (i.e., 2004; $p = 0.065$, 2013; $p = 0.055$, 2014; $p = 0.052$) the probability that the CARB-CAR data were not out of the norm is only $1.87 \times 10^{-4}$%, supporting the null hypothesis rejection. Variation in annual *p*-values is expected according to the number of data points included per year represented as overlapping intervals.

## Howland forest survey protocol comparison

The Howland Research Forest Carbon Project (CAR681) represents the only case in which CARB-CAR and Ameriflux (Ho-1) NEE data are available for the same project location, land area (552 acres; 233.4 hectares), and across shared time intervals (2003–2013) allowing direct comparison of results. Comparability of the Ho-1 NEE1 and CAR681 data is established by the location of two eddy covariance towers within the CARB-CAR survey areas. Half-hourly and integrated annual carbon exchange values recorded at the separate Howland towers were similar, with average annual net carbon uptake differing between the two towers by <6% (*Hollinger et al., 2004*; *Hollinger et al., 2013*; *Richardson et al., 2019*) (see Supplemental Information 10 for project area map, sampling locations and CAR681 webpage offset data; Table 1).

CAR681 methodology and calculations described below are reported in the Project Design Document (PDD, published 12-03-2014) (see Table 1, link to CAR681 project page). Results for CAR681 include data for the period 2008 to 2013 as hindcasted and reported in 2014; a timber inventory was completed in October 2013. The CAR681 Protocol excludes soil carbon (Table 1, FM-6) (*California Air Resources Board, 2015b*) (PDD, pp. 10, 19), employs the common practice method to model onsite above ground carbon stocks of 46 $tCO_2$e $acre^{-1}$ (18.6 $tCO_2$e $hectare^{-1}$) (PDD, p. 33), and established a timber re-inventory interval of 12 years. Model operation (protocol version CARv3.2) included growing the inventory forward 5 years to 2018, then de-growing the inventory to align with the 10/8/2008 start date. Reduced growth rates were selected and employed in model runs to establish the project baseline through a series of estimated growth and harvesting scenarios over 100 years. CAR681 results are available on the CAR website page: Cumulative Performance Report (Table 1, Supplemental Information 10).
**Table 1  CARB-CAR site locations, links to online data source and anomalous features.**

| | Climate action reserve # | Successor climate action reserve # | ARB project ID # | Project name & location | Longitude | Latitude | Functional type | Type of protocol | Acres** | Hectares | Project interval (vintage years) | Project manage-ment | Offsets issued with serial numbers | Cumulative performance report | Anomalous features* | Soil carbon status |
|---|---|---|---|---|---|---|---|---|---|---|---|---|---|---|---|---|
| 1 | CAR1 01 | NA | CAFR0049 | The Van Eck Forest (Humboldt County, CA) | −124.08 | 40.87 | Northern California Coast (Coast Redwood/Douglas Fir Mixed Conifer) & Southern Cascades (Southern Cascade Mixed Conifer) | Early Action | 2,104 | 851 | 2001–2014 | IFM | https://thereserve2.apx.com/mymodule/reg/TabProjectEmissions.asp?id1=101&ad=Prpt&act=update&sBtn=&r=111&Type=PRO&tablename=cr&aProj=pub | https://thereserve2.apx.com/myModule/rpt/myrpt.asp?r=802&md=Prpt&id1=%201011 | ^^(2008) to 2001; ## (2008) to 2014 | %%Reporting Year 2006, Soil Carbon Pool tCO2e = 0 |
| 2 | CAR1 02 | CAR1098 | CAFR0040 | Garcia River Forest (Mendocino, CA) | −123.51 | 38.91 | Temperate coniferous | Early Action | 23,780 | 9,623 | 2005–2017 | IFM | https://thereserve2.apx.com/mymodule/reg/TabProjectEmissions.asp?id1=102&ad=Prpt&act=update&sBtn=&r=111&Type=PRO&tablename=cr&aProj=pub | https://thereserve2.apx.com/myModule/rpt/myrpt.asp?r=802&md=Prpt&id1=%201023 | ^^(2007) 2007–2004; # (2007) 2008–2014 | %%Reporting Year 2010, Soil Carbon Pool Excluded from carbon pool calculations |
| 3 | CAR4 08 | CAR1100 | CAFR0041 | Big River/Salmon Creek Forests (Mendocino, CA) | −123.67 | 39.301 | Temperate coniferous | Early Action | 15,911 | 6,439 | 2007–2017 | IFM | https://thereserve2.apx.com/myModule/rpt/myrpt.asp?r=802&md=Prpt&id1=%20408 | https://thereserve2.apx.com/myModule/rpt/myrpt.asp?r=802&md=Prpt&id1=%20408 | ##(2007) 2007–2014; $2012 | %%2007 Project Submittal Form, Soil Carbon Excluded |
| 4 | CAR4 30 | NA | CAFR0073 | McCloud River (McCloud, CA) | −122 | 41.2 | Temperate broadleaf and mixed | Early Action | 9,200 | 3,723 | 2006–2014 | IFM | https://thereserve2.apx.com/mymodule/reg/TabProjectEmissions.asp?id1=429&ad=Prpt&act=update&sBtn=&r=111&Type=PRO&tablename=cr&aProj=pub | https://thereserve2.apx.com/myModule/rpt/myrpt.asp?r=802&md=Prpt&id1=%20429 | ## (2006) 2006 to 2014 | %%2009 Project Submittal Form, Soil Carbon N/A |
| 5 | CAR4 30 | NA | NA | RPH Ranch (Comptche, Mendocino County, CA) | −123.59 | 39.26 | Temperate coniferous | Early Action | 106 | 43 | 2010 | IFM | https://thereserve2.apx.com/mymodule/reg/TabDocuments.asp?r=111&ad=Prpt&act=update&type=PRO&aProj=pub&tablename=doc&id1=430 | https://thereserve2.apx.com/myModule/rpt/myrpt.asp?r=802&md=Prpt&id1=%20430 | ##(2002) 2002 to 2010; $$(00.00) 2002 to 2009 | %%2008 Project Submittal Form, Soil Carbon N/A |
| 6 | CAR4 97 | NA | CAFR0029 | Blue Source—Alligator River (Hyde County, NC) | −76.031 | 35.631 | Northern Atlantic Coastal Swamp Hardwoods, Cypress | Early Action | 2,272 | 919 | 2010–2017 | AC | https://thereserve2.apx.com/mymodule/reg/TabProjectEmissions.asp?id1=497&ad=Prpt&act=update&sBtn=&r=111&Type=PRO&tablename=cr&aProj=pub | https://thereserve2.apx.com/myModule/rpt/myrpt.asp?r=802&md=Prpt&id1=%20497 | $(2011) | %%Project Design Document, Section 15. Carbon Stock Inventory, Soil Carbon Excluded |

| | Climate action reserve # | Successor climate action reserve # | ARB project ID # | Project name & location | Longitude | Latitude | Functional type | Type of protocol | Acres** | Hectares | Project interval (vintage years) | Project management | Offsets issued with serial numbers | Cumulative performance report | Anomalous features* | Soil carbon status |
|---|---|---|---|---|---|---|---|---|---|---|---|---|---|---|---|---|
| 7 | CAR5 75 | NA | NA | Arcata Sunnybrae Tract (Humbodlt County, CA) | −124.05 | 40.864 | Northern California Coast (Coast Redwood/Douglas Fir Mixed Conifer) & Southern Cascades (Southern Cascade Mixed Conifer) | Not Eligible | 171 | 69 | 2006–2015 | IFM | https://thereserve2.apx.com/mymodule/reg/TabProjectEmissions.asp?id1=575&ad=Prpt&act=update&sBtn=&r=111&Type=PRO&tablename=cr&aProj=pub | https://thereserve2.apx.com/myModule/rpt/myrpt.asp?r=802&md=Prpt&id1=%20575 | ^^(2012) 2012 to 2006; $(2006) one month | %%Project Design Document, 4. On-site Carbon Inventory Methodology |
| 8 | CAR5 82 | CAR1130 | CAFR0103 | Finite Carbon—MWF Brimstone IFM Project I (Scott County, TN) | −84.455 | 36.272 | Mixed Oak | Early Action | 4,861 | 1,967 | 2007–2015 | IFM | https://thereserve2.apx.com/mymodule/reg/TabProjectEmissions.asp?id1=582&ad=Prpt&act=update&sBtn=&r=111&Type=PRO&tablename=cr&aProj=pub | https://thereserve2.apx.com/myModule/rpt/myrpt.asp?r=802&md=Prpt&id1=%20582 | ^^(2013) 2013 to 2007; $(2007) ~3 months; $(2013) ~8 months; $(2015); %(2015) | %%Project Design Document, Item 23. Soil Carbon Pool Absent; Table 23, soil carbon absent, Table 10, soil carbon absent |
| 9 | CAR5 90 | NA | NA | Lompico Forest Carbon Project (Santa Cruz County, CA) | −122.04 | 37.13 | Temperate coniferous Temperate broadleaf, mixed Coastal Redwood forest | Not Eligible | 425 | 172 | 2010–2014 | IFM | https://thereserve2.apx.com/mymodule/reg/TabProjectEmissions.asp?id1=590&ad=Prpt&act=update&sBtn=&r=111&Type=PRO&tablename=cr&aProj=pub | https://thereserve2.apx.com/myModule/rpt/myrpt.asp?r=802&md=Prpt&id1=%20590 | $$(1074.00) 2010 to 2012 | %%Project Design Document, Section 3. Onsite Carbon Inventory Methodology, soil carbon excluded as an optional carbon pool. |
| 10 | CAR6 45 | CAR1088 | CAFR0080 | Finite Carbon—The Forestland Group Champion Property (Franklin, St. Lawrence & Lewis Counties, NY) | −75 | 44.3 | Spruce-fir; Pine and hemlock; Northern hardwoods | Early Action | 100,000 | 40,469 | 2009–2016 | IFM | https://thereserve2.apx.com/mymodule/reg/TabProjectEmissions.asp?id1=645&ad=Prpt&act=update&sBtn=&r=111&Type=PRO&tablename=cr&aProj=pub | https://thereserve2.apx.com/myModule/rpt/myrpt.asp?r=802&md=Prpt&id1=%20645 | ^^(2012) 2012 to 2009; $(2009) ~7 months; $(2014, 2015, 2016); %(2014, 2015, 2016); %(2016) 2015, 2016 | %%Project Design Document, Table 5. Sources, Sinks, and Reservoirs, IFM-6, Soil Carbon excluded |
| 11 | CAR6 46 | NA | NA | Katahdin Iron Works Ecological Reserve (Piscataquis County, ME) | −69.17 | 45.45 | Evergreen Needleleaf Forest | Not Eligible | 10,000 | 4,047 | 2007–2012 | IFM | https://thereserve2.apx.com/mymodule/reg/TabProjectEmissions.asp?r=111&ad=Prpt&act=update&type=PRO&aProj=pub&tablename=cr&id1=646 | https://thereserve2.apx.com/myModule/rpt/myrpt.asp?r=802&md=Prpt&id1=%20646 | $(2007) ~8 months; ^^(2013) 2013 to 2007 | %%Project Design Document, Section 3. Inventory Methodology, IFM-4, Soil Carbon excluded |

**Table 1** (*continued*)

| | Climate action reserve # | Successor climate action reserve # | ARB project ID # | Project name & location | Longitude | Latitude | Functional type | Type of protocol | Acres** | Hectares | Project interval (vintage years) | Project management | Offsets issued with serial numbers | Cumulative performance report | Anomalous features* | Soil carbon status |
|---|---|---|---|---|---|---|---|---|---|---|---|---|---|---|---|---|
| 12 | CAR6 48 | CAR1086 | CAFR0047 | Finite Carbon—Potlatch Moro Big Pine CE (Calhoun County, AR) | −92.54 | 33.5 | Evergreen Needleleaf Forest | Early Action | 16,000 | 6,475 | 2006–2014 | IFM | https://thereserve2.apx.com/mymodule/reg/TabProjectEmissions.asp?id1=648&ad=Prpt&act=update&sBtn=&r=111&Type=PRO&tablename=cr&aProj=pub | https://thereserve2.apx.com/myModule/rpt/myrpt.asp?r=802&md=Prpt&id1=%20648 | $(2006) ~1 month; ^^(2013) 2013 to 2006; $(2013) ~7 months; %(2014) 2012, 2013, 2014; $(2014) ~7 months | %%Project t Design Document, Table 7. Sources, Sinks, and Reservoirs, IFM-6, soil carbon excluded. |
| 13 | CAR6 55 | NA | CAFR0105 | Alder Stream Preserve (Piscataquis County, ME) | −69.015 | 45.114 | Evergreen Needleleaf Forest | Early Action | 1,460 | 591 | 2006–2013 | IFM | https://thereserve2.apx.com/mymodule/reg/TabProjectEmissions.asp?id1=655&ad=Prpt&act=update&sBtn=&r=111&Type=PRO&tablename=cr&aProj=pub | https://thereserve2.apx.com/myModule/rpt/myrpt.asp?r=802&md=Prpt&id1=%20655 | $ initial year 1 month, 2006; ##2012, 2014; ##2006 to 2013 | %%Project Design Document, Section 3, Inventory Methodology, IFM-6, soil carbon excluded |
| 14 | CAR6 57 | CAR1063 | CAFR0002 | Finite Carbon Farm Cove Community Forest Project (Near Grand Lake Stream, Maine) | −67.851 | 45.187 | Evergreen Needleleaf Forest | Early Action | 19,769 | 8,000 | 2010–2015 | IFM | https://thereserve2.apx.com/mymodule/reg/TabProjectEmissions.asp?id1=657&ad=Prpt&act=update&sBtn=&r=111&Type=PRO&tablename=cr&aProj=pub | https://thereserve2.apx.com/myModule/rpt/myrpt.asp?r=802&md=Prpt&id1=%20657 | $(2003) <1 month; ^^(2012) 2011 to 2003 | %%Project Design Document, Section A13. Carbon Pools,. IFM-6, soil carbon excluded |
| 15 | CAR6 58 | CAR1134 | CAFR0087 | Finite Carbon—Brosnan Forest (Near Charleston, SC) | −80.45 | 33.167 | Evergreen Needleleaf Forest | Early Action | 10,209 | 4,131 | 2010–2011; 2015–2016 | IFM | https://thereserve2.apx.com/mymodule/reg/TabProjectEmissions.asp?id1=658&ad=Prpt&act=update&sBtn=&r=111&Type=PRO&tablename=cr&aProj=pub | https://thereserve2.apx.com/myModule/rpt/myrpt.asp?r=802&md=Prpt&id1=%20658 | #(2010) $(2013) ~8 months; ^^(2013) 2011 to 2010 | %%Project Design Document, Section A13. Carbon Pools,. IFM-6, soil carbon excluded |
| 16 | CAR6 59 | NA | CAFR0026 | Blue Source—Pungo River Forest Conservation Project (Washington County, NC) | −76.64 | 35.804 | Atlantic Coastal Plain—Swamp Hardwood and Cypress | Early Action | 704 | 285 | 2003–2016 | AC | https://thereserve2.apx.com/mymodule/reg/TabProjectEmissions.asp?id1=659&ad=Prpt&act=update&sBtn=&r=111&Type=PRO&tablename=cr&aProj=pub | https://thereserve2.apx.com/myModule/rpt/myrpt.asp?r=802&md=Prpt&id1=%20659 | $(2003) <1 month; ^^(2012) 2011 to 2003 | %%Project Design Document, 11.2.3.4.1 Soil carbon was sampled to establish starting carbon stocks that would be degraded if the baseline scenario was followed, e.g., full conversion to agricultural use. The soil carbon was excluded as source of CO2 over the lifetime of the project (e.g., AC-6). |

Marino et al. (2019), *PeerJ*, DOI 10.7717/peerj.7606

**Table 1** (*continued*)

| | Climate action reserve # | Successor climate action reserve # | ARB project ID # | Project name & location | Longitude | Latitude | Functional type | Type of protocol | Acres** | Hectares | Project interval (vintage years) | Project management | Offsets issued with serial numbers | Cumulative performance report | Anomalous features* | Soil carbon status |
|---|---|---|---|---|---|---|---|---|---|---|---|---|---|---|---|---|
| 17 | CAR6 60 | CAR1099 | CAFR0042 | Gualala River Forest (Southern Mendocino County, Near Gualala, CA) | −123.402 | 38.796 | Coastal Redwood and Douglas Fir | Early Action | 13,913 | 5,630 | 2004–2017 | IFM | https://thereserve2.apx.com/mymodule/reg/TabProjectEmissions.asp?id1=660&ad=Prpt&act=update&sBtn=&r=111&Type=PRO&tablename=cr&aProj=pub | https://thereserve2.apx.com/myModule/rpt/myrpt.asp?r=802&md=Prpt&id1=%20660 | ^^(2011) 2010 to 2004; %(2015) 2015, 2016; %(2017) 2016, 2017 | %%Project Design Document, Section 7. Summary of the carbon stock inventory for the Forest Project by each pool, soil carbon pool excluded |
| 18 | CAR6 61 | CAR1140 | CAFR0001 | Willits Woods (Near Willitis, CA) | −123.357 | 39.411 | Coastal Redwood and Douglas Fir | Early Action | 18,008 | 7,288 | 2004–2016 | IFM | https://thereserve2.apx.com/mymodule/reg/TabProjectEmissions.asp?id1=661&ad=Prpt&act=update&sBtn=&r=111&Type=PRO&tablename=cr&aProj=pub | https://thereserve2.apx.com/myModule/rpt/myrpt.asp?r=802&md=Prpt&id1=%20661 | ^^(2011) 2010 to 2004 | %%Project Design Document, Section 5. Calculation methodologies for determining metric tones per acre for each of the included carbon pools, soil carbon excluded |
| 19 | CAR6 72 | NA | CAFR0116 | Hershey Mountain (North of Concord, NH) | −71.667 | 43.567 | Adirondacks & Green Mountains Northern Hardwood | Early Action | 2,141 | 866 | 2007–2013 | IFM | https://thereserve2.apx.com/mymodule/reg/TabProjectEmissions.asp?id1=672&ad=Prpt&act=update&sBtn=&r=111&Type=PRO&tablename=cr&aProj=pub | https://thereserve2.apx.com/myModule/rpt/myrpt.asp?r=802&md=Prpt&id1=%20672 | #(2007); $(2007) ~6 months; ^^(2014) 2013 to 2007 | %%Project Design Document, Section A13. Carbon Pools, soil carbon FM-6 excluded |
| 20 | CAR6 76 | NA | CAFR0031 | Pocosin Lakes Forest Conservation Project (Tyrrell County, NC) | −76.209 | 35.862 | Atlantic Coastal Plain, Swamp Hardwood and Cypress | Early Action | 1,349 | 546 | 2003–2012 | AC | https://thereserve2.apx.com/mymodule/reg/TabProjectEmissions.asp?id1=676&ad=Prpt&act=update&sBtn=&r=111&Type=PRO&tablename=cr&aProj=pub | https://thereserve2.apx.com/myModule/rpt/myrpt.asp?r=802&md=Prpt&id1=%20676 | ^^(2012) 2011 to 2003; $(2003) ~2 months; | %%Project Design Document, 11.2.3.4.1 Soil carbon was sampled to establish starting carbon stocks that would be degraded if the baseline scenario was followed, e.g., full conversion to agricultural use. The soil carbon wasexcluded as source of CO2 over the lifetime of the project (e.g., AC-6) |

Marino et al. (2019), *PeerJ*, DOI 10.7717/peerj.7606

**Table 1** (*continued*)

| | Climate action reserve # | Successor climate action reserve # | ARB project ID # | Project name & location | Longitude | Latitude | Functional type | Type of protocol | Acres** | Hectares | Project interval (vintage years) | Project management | Offsets issued with serial numbers | Cumulative performance report | Anomalous features* | Soil carbon status |
|---|---|---|---|---|---|---|---|---|---|---|---|---|---|---|---|---|
| 21 | CAR6 81 | NA | CAFR0106 | Howland Research Forest (Howland, ME) | −68.627 | 45.246 | Red Spruce and Eastern Hemlock | Early Action | 552 | 223 | 2008–2013 | IFM | https://thereserve2.apx.com/mymodule/reg/TabProjectEmissions.asp?id1=681&ad=Prpt&act=update&sBtn=&r=111&Type=PRO&tablename=cr&aProj=pub | https://thereserve2.apx.com/myModule/rpt/myrpt.asp?r=802&md=Prpt&id1=%20681 | #(2008); $(2008); ^^(2014) 2013 to 2008; %(2008) 2008, 2008 | %%Project Design Document, Section 3 Inventory Methodology, IFM-6, soil carbon excluded |
| 22 | CAR6 83 | NA | CAFR0030 | Francis Beidler Project (Berkeley, Dorchester and Orangeburg Counties, SC) | −80.358 | 33.321 | Native Hardwoods, Softwoods, Mixed Forest | Early Action | 5,548 | 2,245 | 2007–2017 | IFM | https://thereserve2.apx.com/mymodule/reg/TabProjectEmissions.asp?id1=683&ad=Prpt&act=update&sBtn=&r=111&Type=PRO&tablename=cr&aProj=pub | https://thereserve2.apx.com/myModule/rpt/myrpt.asp?r=802&md=Prpt&id1=%20683 | #(2007) ∼5 months; $(2012); ^^(2012) 2012 to 2007; ^^(2015) 2015 to 2012; $(2015); $(2016); $(2017) | %%Project Design Document, Section 3, Inventory Methodology, soil carbon excluded |
| 23 | CAR6 86 | CAR1160 | CAFR0058 | Virginia Conservation Forestry Program—Clifton Farm (Near Rosedale, VA) | −81.86 | 37.022 | Mixed Pine Hardwood, Cove Forests, Oak—Hickory | Early Action | 4,069 | 1,647 | 2004–2016 | IFM | https://thereserve2.apx.com/mymodule/reg/TabProjectEmissions.asp?id1=686&ad=Prpt&act=update&sBtn=&r=111&Type=PRO&tablename=cr&aProj=pub | https://thereserve2.apx.com/myModule/rpt/myrpt.asp?r=802&md=Prpt&id1=%20686 | ^^(2012) 2011 to 2014; ^^(2014) (2013 to 2012); $(2016) 2015, 2016 | %%Project Submittal Form, Item 10, soil carbon excluded |
| 24 | CAR6 88 | NA | CAFR0028 | Blue Source—Noles North Forest Project (Washington and Hyde Counties, NC) | −76.548 | 35.881 | Atlantic Coastal Plain, Swamp Hardwood and Cypress | Early Action | 281 | 114 | 2002–2016 | AC | https://thereserve2.apx.com/mymodule/reg/TabProjectEmissions.asp?id1=688&ad=Prpt&act=update&sBtn=&r=111&Type=PRO&tablename=cr&aProj=pub | https://thereserve2.apx.com/myModule/rpt/myrpt.asp?r=802&md=Prpt&id1=%20688 | $(2002) ∼7 months; ^^(2012) 2011 to 2002; $$(6,099.00) 2003 to 2009; $$(5,830.00) 2013 to 2014 | %%Project Design Document, 11.2.3.4.1 Soil carbon was sampled to establish starting carbon stocks that would be degraded if the baseline scenario was followed, e.g., full conversion to agricultural use. The soil carbon was excluded as source of CO2 over the lifetime of the project (e.g., AC-6) |

**Table 1** (*continued*)

| | Climate action reserve # | Successor climate action reserve # | ARB project ID # | Project name & location | Longitude | Latitude | Functional type | Type of protocol | Acres** | Hectares | Project interval (vintage years) | Project management | Offsets issued with serial numbers | Cumulative performance report | Anomalous features* | Soil carbon status |
|---|---|---|---|---|---|---|---|---|---|---|---|---|---|---|---|---|
| 25 | CAR6 94 | NA | NA | Lucchesi Tract (Humboldt County, CA) | −124.064 | 40.875 | Temperate coniferous, Temperate rainforest; | Not Eligible | 322 | 130 | 2010–2016 | IFM | https://thereserve2.apx.com/mymodule/reg/TabProjectEmissions.asp?id1=694&ad=Prpt&act=update&sBtn=&r=111&Type=PRO&tablename=cr&aProj=pub | https://thereserve2.apx.com/myModule/rpt/myrpt.asp?r=802&md=Prpt&id1=%20694 | ^^(2012) 2011 to 2001; $$(00.00) 2001 to 2009; $$(2,182.00) 2012 to 2016 | %%Project Design Document, Section D. Step 4. Determine the baseline carbon stocks over 100 years for all required and optional carbon pools in the Project Area, soil carbon excluded |
| 26 | CAR6 96 | CAR1159 | CAFR0057 | Rich Mountain (Russell & Washington Counties, NW of Saltville, VA) | −82.03 | 36.831 | Allegheny & North Cumberland Mountains—Mixed Pine Hardwood, Cove Forests, Northern Hardwoods, Oak—Hickory | Early Action | 5,750 | 2,327 | 2002–2016 | IFM | https://thereserve2.apx.com/mymodule/reg/TabProjectEmissions.asp?id1=696&ad=Prpt&act=update&sBtn=&r=111&Type=PRO&tablename=cr&aProj=pub | https://thereserve2.apx.com/myModule/rpt/myrpt.asp?r=802&md=Prpt&id1=%20696 | $(2002) ~6 months; ^^(2012) 2011 to 2002; ^^(2015) 2014 to 2013; %(2016) 2015, 2016 | %%Project Submittal Form, item 10. IFM-6, soil carbon excluded |
| 27 | CAR6 97 | CAR1147 | CAFR0102 | Tazewell—Elk Garden (Russell, Washington, and Tazewell Co. near Tazewell, VA) | −81.559 | 37.124 | Allegheny & North Cumberland Mountains—Mixed Pine Hardwood, Cove Forests, Northern Hardwoods, Oak—Hickory | Early Action | 11,697 | 4,734 | 2007–2016 | IFM | https://thereserve2.apx.com/mymodule/reg/TabProjectEmissions.asp?id1=697&ad=Prpt&act=update&sBtn=&r=111&Type=PRO&tablename=cr&aProj=pub | https://thereserve2.apx.com/myModule/rpt/myrpt.asp?r=802&md=Prpt&id1=%20697 | ^^(2014) 2013 to 2005; %(2014) 2015, 2016; %(2016) 2015, 2016 | %%Project Submittal Form, item 10. IFM-6, soil carbon excluded |
| 28 | CAR7 30 | CAR1139 | CAFR0123 | Usal Redwood Forest (Mendocino County, CA) | −123.847 | 39.876 | Coast Redwood/Douglas-fir Mixed Conifer | Early Action | 49,000 | 19,830 | 2007–2017 | IFM | https://thereserve2.apx.com/mymodule/reg/TabProjectEmissions.asp?id1=730&ad=Prpt&act=update&sBtn=&r=111&Type=PRO&tablename=cr&aProj=pub | https://thereserve2.apx.com/myModule/rpt/myrpt.asp?r=802&md=Prpt&id1=%20730 | #(2007); $(2207) ~6 months; ^^(2015) 2015 to 2007 | %%Project Submittal Form, item 10. IFM-6, soil carbon excluded |
| 29 | CAR7 49 | CAR1109 | CAFR0063 | Green Assets—Middleton (Charleston, SC) | −80.141 | 32.9 | SE Middle Mixed Forest Piedmont Atlantic Coastal Plain & Flatwoods | Early Action | 3,732 | 1,510 | 2007–2017 | AC | https://thereserve2.apx.com/mymodule/reg/TabProjectEmissions.asp?id1=749&ad=Prpt&act=update&sBtn=&r=111&Type=PRO&tablename=cr&aProj=pub | https://thereserve2.apx.com/myModule/rpt/myrpt.asp?r=802&md=Prpt&id1=%20749 | $(2007) <1 month; ^^(2013) 2011 to 2007; $(2011) 2013, 2014; ^^(2014) 2013 to 2011; %(2015) 2014, 2015; %(2016) 2015, 2016; %(2017) 2016, 2017 | %%Project Design Document, Section 11.2.3 Data gathering procedures and parameters, AC-6, soil carbon excluded, Table 5, soil carbon emissions excluded |

Peer J

**Table 1** (*continued*)

| Climate action reserve # | Successor climate action reserve # | ARB project ID # | Project name & location | Longitude | Latitude | Functional type | Type of protocol | Acres** | Hectares | Project interval (vintage years) | Project manage-ment | Offsets issued with serial numbers | Cumulative performance report | Anomalous features* | Soil carbon status |
|---|---|---|---|---|---|---|---|---|---|---|---|---|---|---|---|
| 30 | CAR7 77 | NA | CAFR0064 | Yurok Tribe Sustainable Forest Project (Northwest Humboldt County, CA) | −123.8 | 41.406 | Northern California Coast (Coast Redwood/Douglas Fir Mixed Conifer) & Southern Cascades (Southern Cascade Mixed Conifer) | Early Action | 21,240 | 8,596 | 2011–2014 | IFM | https://thereserve2.apx.com/mymodule/reg/TabProjectEmissions.asp?id1=777&ad=Prpt&act=update&sBtn=&r=111&Type=PRO&tablename=cr&aProj=pub | https://thereserve2.apx.com/myModule/rpt/myrpt.asp?r=802&md=Prpt&id1=%20777 | #(2011) $(2011) ~8 months; ^^(2014) 2013 to 2012 | %%Project Desing Document, Section 3. Inventory Methodology, IFM-6, soil carbon excluded |
| 31 | CAR7 80 | CAR1062 | CAFR0088 | Shannondale Tree Farm (Washington County, NC) | −91.45 | 37.367 | Atlantic Coastal Plain—Atlantic Coastal Plain Swamp Hardwood and Cypress | Early Action | 4037 | 1,634 | 2010–2013 | AC | https://thereserve2.apx.com/mymodule/reg/TabProjectEmissions.asp?id1=780&ad=Prpt&act=update&sBtn=&r=111&Type=PRO&tablename=cr&aProj=pub | https://thereserve2.apx.com/myModule/rpt/myrpt.asp?r=802&md=Prpt&id1=%20780 | #(2013); $(2013) ~4 months; ^^(2013) 2011 to 2010; ^^(2015) 2013 to 2012 | %%Project Design Document, Section A13. Carbon Pools, IFM-6, soil carbon excluded |
| 32 | CAR8 02 | NA | CAFR0027 | Noles South Forest Project (Washington County, NC) | −76.548 | 35.865 | Atlantic Coastal Plain—Atlantic Coastal Plain Swamp Hardwood and Cypress | Early Action | 324 | 131 | 2003–2016 | AC | https://thereserve2.apx.com/mymodule/reg/TabProjectEmissions.asp?id1=802&ad=Prpt&act=update&sBtn=&r=111&Type=PRO&tablename=cr&aProj=pub | https://thereserve2.apx.com/myModule/rpt/myrpt.asp?r=802&md=Prpt&id1=%20802 | $(2003) ~1 month; ^^(2012) 2011 to 2003; $$(5,180.00) 2005 to 2009; $$(5,830.00) 2011, 2012 | %%Project Design Document, 11.2.3.4.1 Soil carbon was sampled to establish starting carbon stocks that would be degraded if the baseline scenario was followed, e.g., full conversion to agricultural use. The soil carbon was excluded as source of CO2 over the lifetime of the project (e.g., AC-6) |
| 33 | CAR9 35 | NA | NA | Arcata City Barnum Tract (Arcata, CA) | −124.049 | 40.876 | Northern California Coast Redwood/Douglas-fir Mixed Conifer | Not Eligible | 280 | 113 | 2003–2016 | IFM | https://thereserve2.apx.com/mymodule/reg/TabProjectEmissions.asp?id1=935&ad=Prpt&act=update&sBtn=&r=111&Type=PRO&tablename=cr&aProj=pub | https://thereserve2.apx.com/myModule/rpt/myrpt.asp?r=802&md=Prpt&id1=%20935 | $(2003) ~11 months; ^^(2012) 2011 to 2003; $$(2,904.00) 2005 to 2011; $$(2,527.00) 2012 to 2016 | %%Project Design Document, Section Step 4. Determine the baseline carbon stocks over 100 years for all required and optional carbon pools in the Project Area, IFM-6. soil carbon excluded |

| | Climate action reserve # | Successor climate action reserve # | ARB project ID # | Project name & location | Longitude | Latitude | Functional type | Type of protocol | Acres** | Hectares | Project interval (vintage years) | Project management | Offsets issued with serial numbers | Cumulative performance report | Anomalous features* | Soil carbon status |
|---|---|---|---|---|---|---|---|---|---|---|---|---|---|---|---|---|
| 34 | CAR1 013 | NA | CAFR5055 | Buckeye Forest Project (Sonoma County, CA) | −123.31 | 38.74 | Coast Redwood/Douglas-fir Mixed Conifer and Northern California Coast Mixed Oak Woodland | Compliance | 19,525 | 7,901 | 2014–2017 | IFM | https://thereserve2.apx.com/mymodule/reg/TabProjectEmissions.asp?id1=1013&ad=Prpt&act=update&sBtn=&r=111&Type=PRO&tablename=cr&aProj=pub | NA | #(2014); $%(2014); $%(2015); $%(2016); $%(2017) | %%Application for Listing, Part VII, Carbon Stock Inventory, IMF-6, Not applicable |
| 35 | CAR1 015 | NA | CAFR0100 | Rips Redwoods (Sonoma County, CA) | −123.212 | 38.711 | Coast Redwood/Douglas-fir Mixed Conifer and Northern California Coast Mixed Oak Woodland | Early Action | 1426 | 577 | 2013–2014 | IFM | https://thereserve2.apx.com/mymodule/reg/TabProjectEmissions.asp?id1=1015&ad=Prpt&act=update&sBtn=&r=111&Type=PRO&tablename=cr&aProj=pub | https://thereserve2.apx.com/myModule/rpt/myrpt.asp?r=802&md=Prpt&id1=%201015 | #(2013); $(2013) ~7 months | %%Project Design Document, Section 2.B.B. Carbon Sinks, Sources and Reservoirs, IMF-6, absent |
| 36 | CAR1 032 | NA | CAFR5037 | Virginia Highlands I (Russell, Buchanan and Dickenson Counties, VA) | −82.347 | 37.085 | oak-hickory, loblolly-shortleaf pine, and mixed oak-pine | Compliance | 9,753 | 3,947 | 2013 | IFM | https://thereserve2.apx.com/mymodule/reg/TabProjectEmissions.asp?id1=1032&ad=Prpt&act=update&sBtn=&r=111&Type=PRO&tablename=cr&aProj=pub | NA | #(2013); $(2013) ~7 months | %%Application for Listing, Part VII, Carbon Stock Inventory, IMF-6, Not applicable |
| 37 | CAR1 041 | NA | CAFR5038 | Sacramento Canyon ARB001 (Shasta County, CA) | −122.29 | 41.05 | Southern Cascade, Mixed Conifer | Compliance | 16,941 | 6,856 | 2015–2017 | IFM | https://thereserve2.apx.com/mymodule/reg/TabProjectEmissions.asp?id1=1041&ad=Prpt&act=update&sBtn=&r=111&Type=PRO&tablename=cr&aProj=pub | NA | #(2015); %(2015) 2013, 2014, 2015; $(2016) 2015, 2016; $(2017) 2016, 2017 | %%Application for Listing, Part VII, Carbon Stock Inventory, IMF-6, Not applicable |
| 38 | CAR1 046 | NA | CAFR5076 | Trinity Timberlands University Hill Project (Trinity County, CA) | −123.5 | 40.58 | "Northern California Coast (Coast Redwood/Douglas Fir Mixed Conifer) & Southern Cascades (Southern Cascade Mixed Conifer)" | Compliance | 11,900 | 4,816 | 2014 | IFM | https://thereserve2.apx.com/mymodule/reg/TabProjectEmissions.asp?id1=1046&ad=Prpt&act=update&sBtn=&r=111&Type=PRO&tablename=cr&aProj=pub | NA | #(2014); $(2013) ~10 months; %(2013) 2013, 2014 | %%Application for Listing, Part VII, Carbon Stock Inventory, IMF-6, Not applicable (Attachment E) |

Peer J

| | Climate action reserve # | Successor climate action reserve # | ARB project ID # | Project name & location | Longitude | Latitude | Functional type | Type of protocol | Acres** | Hectares | Project interval (vintage years) | Project manage-ment | Offsets issued with serial numbers | Cumulative per-formance report | Anomalous features* | Soil carbon status |
|---|---|---|---|---|---|---|---|---|---|---|---|---|---|---|---|---|
| 39 | CAR1 066 | NA | CAFR5058 | Buck Mountain ARB002 (Siskiyou County, CA) | −121.85 | 41.38 | Southern Cascade, Mixed Conifer | Compliance | 12,486 | 5,053 | 2015–2017 | IFM | https://thereserve2.apx.com/mymodule/reg/TabProjectEmissions.asp?id1=1066&ad=Prpt&act=update&sBtn=&r=111&Type=PRO&tablename=cr&aProj=pub | NA | #(2015); $(2015) ~9 months; %(2015) 2014, 2015; $(2016) 2015, 2016; $(2017) 2016, 2017 | %%Application for Listing, Part VII, Carbon Stock Inventory, IMF-6, Not applicable |
| 40 | CAR1 067 | NA | CAFR5063 | Sustainable Mountain (Humboldt County, CA) (near Willow Creek) | −123.76 | 40.91 | Douglas Fir Mixed Conifer | Compliance | 2,112 | 855 | 2015 | IFM | https://thereserve2.apx.com/mymodule/reg/TabProjectEmissions.asp?id1=1067&ad=Prpt&act=update&sBtn=&r=111&Type=PRO&tablename=cr&aProj=pub | NA | $(2014) ~6 months; %(2014) 2013, 2014 | %%Application for Listing, Part VII, Carbon Stock Inventory, IMF-6, excluded |
| 41 | CAR1 092 | NA | CAFR5087 | Big Valley (Near Aiden, CA) | −121.24 | 41.13 | Douglas Fir Mixed Conifer | Active | 14,622 | 5,917 | 2016 | IFM | https://thereserve2.apx.com/mymodule/reg/TabProjectEmissions.asp?id1=1092&ad=Prpt&act=update&sBtn=&r=111&Type=PRO&tablename=cr&aProj=pub | NA | #(2016); %(2016) 2014, 2015, 2016 | %%Application for Listing, Part VII, Carbon Stock Inventory, IMF-6, Not applicable |
| 42 | CAR1 094 | NA | CAFR5095 | Ashford III (Ashford, WA) | −122.04 | 46.46 | Northwest Cascade Mixed Conifer | Compliance | 5,290 | 2,141 | 2014 | IFM | https://thereserve2.apx.com/mymodule/reg/TabProjectEmissions.asp?id1=1094&ad=Prpt&act=update&sBtn=&r=111&Type=PRO&tablename=cr&aProj=pub | NA | #(2014); %(2014) 2012, 2013, 2014 | %%Application for Listing, Part VII, Carbon Stock Inventory, IMF-6, Not applicable |
| 43 | CAR1 095 | NA | CAFR5096 | Brushy Mountain (Mendo-cino County, CA) | −123.26 | 39.63 | Southern Cascade Mixed Conifer | Compliance | 16,392 | 6,634 | 2014-2017 | IFM | https://thereserve2.apx.com/mymodule/reg/TabProjectEmissions.asp?id1=1095&ad=Prpt&act=update&sBtn=&r=111&Type=PRO&tablename=cr&aProj=pub | NA | #(2014); $(2014) ~8 months; $(2016) 2015, 2016; %(2015) 2014, 2015; %(2016) 2015, 2016; %(2017) 2016, 2017; | %%Application for Listing, Part VII, Carbon Stock Inventory, IMF-6, excluded (Addendum to Listing Application) |

**Table 1** (*continued*)

| | Climate action reserve # | Successor climate action reserve # | ARB project ID # | Project name & location | Longitude | Latitude | Functional type | Type of protocol | Acres** | Hectares | Project interval (vintage years) | Project manage-ment | Offsets issued with serial numbers | Cumulative per-formance report | Anomalous features* | Soil carbon status |
|---|---|---|---|---|---|---|---|---|---|---|---|---|---|---|---|---|
| 44 | CAR1 102 | NA | CAFR5148 | Montesol Forest Carbon (Napa and Lake County, CA) | −122.564 | 38.671 | Southern Cascade Mixed Conifer | Compliance | 3,102 | 1,255 | 2016 | IFM | https://thereserve2.apx.com/mymodule/reg/TabProjectEmissions.asp?id1=1102&ad=Prpt&act=update&sBtn=&r=111&Type=PRO&tablename=cr&aProj=pub | NA | #(2016); $(2016) ~7 months | %%Application for Listing, Part VII, Carbon Stock Inventory, IMF-6, Not applicable |
| 45 | CAR1 103 | NA | CAFR5149 | Forest Carbon Partners—Glass Ranch Improved Forest Management Project (Humboldt County, CA) | −123.644 | 40.346 | Southern Cascade Mixed Conifer | Compliance | 22,676 | 9,177 | 2015 | IFM | https://thereserve2.apx.com/mymodule/reg/TabProjectEmissions.asp?id1=1103&ad=Prpt&act=update&sBtn=&r=111&Type=PRO&tablename=cr&aProj=pub | NA | #(2015); %(2015) 2014, 2015 | %%Application for Listing, Part VII, Carbon Stock Inventory, IMF-6, Not applicable |
| 46 | CAR1 104 | NA | CAFR5150 | Forest Carbon Partners—Gabrych Ranch Project (Humboldt County and Trinity County, CA) | −123.606 | 40.713 | Southern Cascade Mixed Conifer | Compliance | 4,039 | 1,635 | 2015 | IFM | https://thereserve2.apx.com/mymodule/reg/TabProjectEmissions.asp?id1=1104&ad=Prpt&act=update&sBtn=&r=111&Type=PRO&tablename=cr&aProj=pub | NA | #(2015); %(2015) 2014, 2015 | %%Application for Listing, Part VII, Carbon Stock Inventory, IMF-6, Not applicable |
| 47 | CAR1 114 | NA | CAFR5114 | Crane Valley | −123.606 | 40.713 | Southern Cascade Mixed Oak Woodland and Sierra Mixed Oak Woodland | Compliance | 19,384 | 7,844 | 2016 | IFM | https://thereserve2.apx.com/mymodule/reg/TabProjectEmissions.asp?r=111&ad=Prpt&act=update&type=PRO&aProj=pub&tablename=cr&id1=1114 | NA | #(2016); %(2016) 2014, 2015, 2016 | %%PART VII. CARBON STOCK INVENTORY IFM-6 Soil (if applicable): N/A |
| 48 | CAR1 175 | NA | CAFR5195 | Finite Carbon—Passamaquoddy Tribe (Frankin, Somerset, Penobscot, Hancock, and Washington Counties, ME) | −67.63 | 45.288 | New Brunswick Foothills & Lowlands, White Mountains Mixed Hardwoods | Compliance | 98,492 | 39,858 | 2015–2016 | IFM | https://thereserve2.apx.com/mymodule/reg/TabProjectEmissions.asp?id1=1175&ad=Prpt&act=update&sBtn=&r=111&Type=PRO&tablename=cr&aProj=pub | NA | #(2015); %(2015) 2014, 2015; %(2016) 2015, 2016; %(2016) 2015, 2016 | %%PART VII. CARBON STOCK INVENTORY IFM-6 Soil (if applicable): Excluded |

Marino et al. (2019), *PeerJ*, DOI 10.7717/peerj.7606

**Table 1** (*continued*)

| | Climate action reserve # | Successor climate action reserve # | ARB project ID # | Project name & location | Longitude | Latitude | Functional type | Type of protocol | Acres** | Hectares | Project interval (vintage years) | Project management | Offsets issued with serial numbers | Cumulative performance report | Anomalous features* | Soil carbon status |
|---|---|---|---|---|---|---|---|---|---|---|---|---|---|---|---|---|
| 49 | CAR1 180 | NA | CAFR5280 | Maillard Ranch (Mendocino County, CA) | −123.36 | 39.92 | Temperate coniferous | Compliance | 12,360 | 5,002 | 2016 | IFM | https://thereserve2.apx.com/mymodule/reg/TabProjectEmissions.asp?id1=1180&ad=Prpt&act=update&sBtn=&r=111&Type=PRO&tablename=cr&aProj=pub | NA | #(2016); %(2016) 2015, 2016 | %%PART VII. CARBON STOCK INVENTORY IFM-6 Soil (if applicable): Excluded |
| 50 | CAR1 183 | NA | CAFR5283 | Forest Carbon Partners-Mescalero Apache Tribe (Otero & Lincoln County, NM) | −105.65 | 33.17 | Red Spruce and Eastern Hemlock | Compliance | 221,822 | 89,768 | 2016 | IFM | https://thereserve2.apx.com/mymodule/reg/TabProjectEmissions.asp?r=111&ad=Prpt&act=update&type=PRO&aProj=pub&tablename=cr&id1=1183 | NA | #(2016); $(2016) ~10 months; %(2016) 2015, 2016 | %%PART VII. CARBON STOCK INVENTORY IFM-6 Soil (if applicable):Not Applicable |
| 51 | CAR1 191 | NA | CAFR5291 | Hollow Tree (Mendocino County, CA) | −123.782 | 39.85 | Coast Redwood/Douglas-fir Mixed Conifer | Compliance | 20,295 | 8,213 | 2016 | IFM | https://thereserve2.apx.com/mymodule/reg/TabDocuments.asp?r=111&ad=Prpt&act=update&type=PRO&aProj=pub&tablename=doc&id1=1191 | NA | #(2016); %(2016) 2015, 2016 | %%PART VII. CARBON STOCK INVENTORY IFM-6 Soil (if applicable): Not Applicable |
| 52 | CAR1 197 | NA | CAFR5297 | Upper Hudson Woodlands ATP, LP (Warren, Hamilton, Essex, Washington, Saratoga and Fulton, NY) | −74.33 | 43.88 | Mixed conifer-/mixed hardwood forest | Compliance | 86,825 | 35,137 | 2017 | IFM | https://thereserve2.apx.com/mymodule/reg/TabProjectEmissions.asp?id1=1197&ad=Prpt&act=update&sBtn=&r=111&Type=PRO&tablename=cr&aProj=pub | NA | #(2017); %(2017) 2015, 2016, 2017 | %%PART VII. CARBON STOCK INVENTORY IFM-6 Soil (if applicable): Excluded |
| 53 | CAR1 204 | NA | CAFR5304 | AMC Silver Lake (Piscataquis & Aroostook Counties, ME) | −69.15 | 45.44 | Spruce-Fir and Mixed Hardwood forests | Compliance | 89,315 | 36,145 | 2017 | IFM | https://thereserve2.apx.com/mymodule/reg/TabProjectEmissions.asp?id1=1204&ad=Prpt&act=update&sBtn=&r=111&Type=PRO&tablename=cr&aProj=pub | NA | #(2017); %(2017) 2015, 2016, 2017 | %%PART VII. CARBON STOCK INVENTORY IFM-6 Soil (if applicable): Excluded |

**Table 1** (*continued*)

| | Climate action reserve # | Successor climate action reserve # | ARB project ID # | Project name & location | Longitude | Latitude | Functional type | Type of protocol | Acres** | Hectares | Project interval (vintage years) | Project manage-ment | Offsets issued with serial numbers | Cumulative per-formance report | Anomalous features* | Soil carbon status |
|---|---|---|---|---|---|---|---|---|---|---|---|---|---|---|---|---|
| 54 | CAR1 209 | NA | CAFR5309 | Wolf River (Antigo, WI) | −88.86 | 45.23 | Northern hard-wood/mixed conifer forest-land | Compliance | 17,722 | 7,172 | 2017 | IFM | https://thereserve2. apx.com/ mymodule/reg/ TabProjectEmissions. asp?r=111&ad= Prpt&act=update& type=PRO&aProj= pub&tablename= cr&id1=1209 | NA | #(2017); %(2017) 2015, 2016, 2017 | %%PART VII. CARBON STOCK IN-VENTORY IFM-6 Soil (if applicable): Excluded |
| 55 | CAR1 213 | NA | CAFR5313 | MWF Adirondacks (Franklin, St. Lawrence & Lewis Coun-ties, NY) | −74.91 | 44.35 | Adirondacks & Green Moun-tains Northern Hardwood | Compliance | 100,094 | 40,507 | 2017 | IFM | https://thereserve2. apx.com/ mymodule/reg/ TabProjectEmissions. asp?r=111&ad= Prpt&act=update& type=PRO&aProj= pub&tablename= cr&id1=1213 | NA | #(2017); %(2017) 2015, 2016, 2017 | %%PART VII. CARBON STOCK IN-VENTORY IFM-6 Soil (if applicable): Excluded |
| 56 | CAR1 215 | NA | CAFR5315 | Molpus Ataya (Campbell & Claiborne Counties, TN) | −83.89 | 36.54 | Allegheny & North Cumberland Mountains— Mixed Pine Hardwood, Cove Forests, North-ern Hardwoods, Oak—Hickory | Compliance | 26,261 | 10,627 | 2017 | IFM | https://thereserve2. apx.com/ mymodule/reg/ TabProjectEmissions. asp?r=111&ad= Prpt&act=update& type=PRO&aProj= pub&tablename= cr&id1=1215 | NA | #(2017); %(2017) 2015, 2016, 2017 | %%PART VII. CARBON STOCK IN-VENTORY IFM-6 Soil (if applicable): Excluded |
| 57 | CAR1 217 | NA | CAFR5317 | West Grand Lake (Wash-ington County, ME) | −67.75 | 45.23 | New Brunswick Foothills & Low-lands, White Mountains Mixed Hard-woods | Compliance | 19,552 | 7,912 | 2015 | IFM | https://thereserve2. apx.com/ mymodule/reg/ TabProjectEmissions. asp?r=111&ad= Prpt&act=update& type=PRO&aProj= pub&tablename= cr&id1=1217 | NA | #(2015); %(2015) 2013, 2014, 2015 | %%PART VII. CARBON STOCK IN-VENTORY IFM-6 Soil (if applicable): Excluded |

**Table 1** (*continued*)

| | Climate action reserve # | Successor climate action reserve # | ARB project ID # | Project name & location | Longitude | Latitude | Functional type | Type of protocol | Acres** | Hectares | Project interval (vintage years) | Project management | Offsets issued with serial numbers | Cumulative performance report | Anomalous features* | Soil carbon status |
|---|---|---|---|---|---|---|---|---|---|---|---|---|---|---|---|---|
| 58 | CAR9 73 | NA | CAFR5003 | Bishop Project (Near Bessemer, MI, and other locations) | −87.852 | 46.5620 | Tree cover, broadleaved, deciduous, closed to open (>15%) | NA | 2,112.86 | 855.044862 | 2013–2016 | IFM | https://thereserve2.apx.com/mymodule/reg/TabProjectEmissions.asp?id1=973&ad=Prpt&act=update&sBtn=&r=111&Type=PRO&tablename=cr&aProj=pub | https://thereserve2.apx.com/myModule/rpt/myrpt.asp?r=802&md=Prpt&id1=%201004 | $2013 | %% Blue Source—Bishop Improved Forest Management Project ARB Project Listing Form Attachments February 4, 2013, Part V.B, Soil carbon excluded |
| 59 | CAR1 004 | NA | NA | Berry Summit (Near Eureka, CA) | −123.758 | 40.905 | Tree cover, needle leaved, evergreen, closed to open (>15%) | NA | 2,112.86 | | 2013 | IFM | https://thereserve2.apx.com/mymodule/reg/TabProjectEmissions.asp?id1=1004&ad=Prpt&act=update&sBtn=&r=111&Type=PRO&tablename=cr&aProj=pub | NA | $2013 | %% Project Description Document, Table 5, IFM-6 not included |
| 60 | CAR1 174 | NA | CAFR5224 | Eddie Ranch (Mendocino County, CA) | −123.17 | 39.456 | Tree cover, needle leaved, evergreen, closed to open (>15%) | NA | 2,286 | | 2017 | IFM | https://thereserve2.apx.com/mymodule/reg/TabProjectEmissions.asp?id1=1174&ad=Prpt&act=update&sBtn=&r=111&Type=PRO&tablename=cr&aProj=pub | NA | $2017, %2017 | %% Application for Listing, Part VII-A, IFM-6, Soil Carbon, not applicable |
| 61 | CAR1 190 | NA | CAFR5220 | Greenwood Creek (Mendocino County, CA) | −123.631 | 39.073 | Tree cover, needle leaved, evergreen, closed to open (>15%) | NA | 8,659 | 3,594.17 | 2015–2017 | IFM | https://thereserve2.apx.com/mymodule/reg/TabProjectEmissions.asp?id1=1190&ad=Prpt&act=update&sBtn=&r=111&Type=PRO&tablename=cr&aProj=pub | NA | $2015-2016-2017; %2015, 2016, 2017 | %% Application for Listing, Part II-C, IFM-6, Soil Carbon, not applicable |
| 62 | CAR1 262 | NA | NA | San Juan Lachao Pueblo Nuevo, Oaxaca, Mexico | −97.125 | 16.158 | Tree Cover, broadleaved, deciduous, closed | NA | 32,840.31 | 13,290 | 2014–2016 | Forestry | https://thereserve2.apx.com/mymodule/reg/TabProjectEmissions.asp?id1=1262&ad=Prpt&act=update&sBtn=&r=111&Type=PRO&tablename=cr&aProj=pub | NA | $2014, 2015, 2016; %2014, 2015 | %% Carbono en el suelo: No se incluye, REPORTE DE PROYECTO Captura de Carbono en San Juan Lachao, Oaxaca San Juan Lachao Pueblo Nuevo, Oaxaca 11 de octubre de 2017 CAR1262 |

**Table 1** (*continued*)

| Climate action reserve # | Successor climate action reserve # | ARB project ID # | Project name & location | Longitude | Latitude | Functional type | Type of protocol | Acres** | Hectares | Project interval (vintage years) | Project manage-ment | Offsets issued with serial numbers | Cumulative per-formance report | Anomalous features* | Soil carbon status |
|---|---|---|---|---|---|---|---|---|---|---|---|---|---|---|---|
| 63 | CAR1 306 | NA | NA | Ejido San Nicolás Toto-lapan, CDMX, Mexico | −99.2544 | 19.2994 | Tree cover, broadleaved, de-ciduous, closed to open (>15%) | NA | 5,302.83 | 2,145.98 | 2017–2018 | Forestry | https://thereserve2.apx.com/mymodule/reg/TabProjectEmissions.asp?id1=1306&ad=Prpt&act=update&sBtn=&r=111&Type=PRO&tablename=cr&aProj=pub | NA | $2017, 2018 | %% https://thereserve2.apx.com/mymodule/reg/TabDocuments.asp?r=111&ad=Prpt&act=update&type=PRO&aProj=pub&tablename=doc&id1=1306 |

**Notes.**

*Anomalous Features

# Vintage year is an outlier defined in Fig. 1A. Issue date may vary.

^^Backward model reporting year run (in parenthesis) for the interval noted. Issue date may vary.

## Forward model reporting year run (in parenthesis) for the interval noted. Issue date may vary.

$ A single vintage year (in parenthesis) is reported as a partial year or a single vintage year is split representing two or more reported carbon sequestration intervals as indicated. Issue date may vary.

$$ Exact values (in parenthesis) for carbon sequestration are repeated over interval as indicated. Issue date may vary.

% A single vintage year (in parenthesis) represents two or more reported years and or multiple net carbon sequestration years as indicated. Issue date may vary.

**Project size ranged from 221,822 to 106 acres with a mean size of 21,256 acres, standard deviation of 37,451 acres. Issue date may vary.

%% Soil carbon pool excluded and not directly measured as specified in project documentation.

IFM, Improved Forest Management. This protocol applies to forest offset projects that involve management activities that maintain or increase carbon stocks on forested land relative to baseline levels of carbon stocks.

AC, Avoided Conversion. This protocol applies to forest offset projects that involve preventing the conversion of forestland to a non-forest land use by dedicating the land to continuous forest cover through a qualified conservation easement or transfer to public ownership, excluding transfer to federal ownership.

Marino et al. (2019), *PeerJ*, DOI 10.7717/peerj.7606

**Table 2 Results of a hypothesis test with a null hypothesis that the difference between the CARB-CAR and the NEE1 means is under the allowed 5% threshold.** The test is performed separately for all years between 2002 and 2014, the *p*-values are recorded in the last two rows.

| | 2002 | 2003 | 2004 | 2005 | 2006 | 2007 | 2008 | 2009 | 2010 | 2011 | 2012 | 2013 | 2014 | 2015 |
|---|---|---|---|---|---|---|---|---|---|---|---|---|---|---|
| CARB-CAR mean | −742.31 | −719.94 | −539.51 | −1021.69 | −2038.03 | −992.42 | −950.47 | −593.86 | −983.74 | −725.04 | −682.77 | −865.27 | −999.38 | −1432.69 |
| CARB-CAR SD | 928.79 | 585.94 | 1049.34 | 683.21 | 3433.06 | 1418.73 | 1170.20 | 644.84 | 1460.80 | 1190.51 | 1049.74 | 1541.08 | 1590.56 | 1835.66 |
| CARB-CAR (n) | 2.00 | 5.00 | 9.00 | 9.00 | 12.00 | 23.00 | 24.00 | 25.00 | 32.00 | 32.00 | 31.00 | 32.00 | 30.00 | 23.00 |
| NEE1 mean | −190.55 | −189.15 | −243.45 | −267.26 | −225.08 | −241.51 | −217.16 | −206.75 | −184.51 | −92.29 | −93.45 | −53.87 | −2.38 | −9.29 |
| NEE1 SD | 249.06 | 266.80 | 268.25 | 250.02 | 243.39 | 237.56 | 254.86 | 275.89 | 244.93 | 231.71 | 161.39 | 199.85 | 214.46 | 207.25 |
| NEE1 (n) | 40.00 | 48.00 | 45.00 | 42.00 | 44.00 | 42.00 | 41.00 | 31.00 | 24.00 | 17.00 | 12.00 | 11.00 | 8.00 | 7.00 |
| *p*-value | 0.01 | 0.00 | 0.07 | 0.00 | 0.00 | 0.00 | 0.00 | 0.00 | 0.01 | 0.02 | 0.04 | 0.05 | 0.05 | 0.03 |
| *p*-value in % | 0.87 | 0.05 | 6.53 | 0.00 | 0.07 | 0.13 | 0.03 | 0.36 | 0.81 | 2.37 | 3.88 | 5.48 | 5.24 | 3.25 |

Carbon sequestration for the initial CAR681 vintage year, 2008, is reported as $-43{,}687.0$ $tCO_2e$, or $-5{,}334.7$ gC $m^{-2}yr^{-1}$, $\sim26\times$ in excess of the reported NEE1 population mean (e.g., $-198.0$ gC $m^{-2}$ $yr^{-2}$) and $\sim34\times$ the NEE2 population mean (e.g., $-156$ gC $m^{-2}$ $yr^{-2}$) (Supplemental Information 9, filled rectangle symbol). Subsequent vintage years, 2009 to 2013, were invariant with exact values of $-1{,}033.00$ $tCO_2e$, or $-126.1$ gC $m^{-2}yr^{-1}$. CAR681 data for the interval 2008 to 2013 yield a mean and standard deviation of $-994.0 \pm 2{,}126.1$ gC $m^{-2}yr^{-1}$, respectively. The total CAR681 offsets issued equal 48,852 $tCO_2e$ (Table 1, Supplemental Information 10, CAR681 Project Page, Cumulative Performance Report). CAR681 results were audited and approved (see link, Table 1). Serial numbers were assigned to offsets on 03/13/2015 (Table 1, Supplement S8).

In contrast, Ho-1 NEE for 2008 was reported as $-287.1$ gC $m^{-2}yr^{-1}$ (*Hollinger et al., 2013*; *Hollinger et al., 2016*), $\sim19\times$ smaller compared to the 2008 CAR681 value, representing an over-estimation error of $\sim1{,}757\%$ (e.g., footnote 2). The CAR681 result clearly exceeds the 5% CAR invalidation threshold criteria and is arguably invalid. Ho-1 NEE values for the years 2009 to 2013 ranged from $-191.9$ to $-330.9$ gC $m^{-2}yr^{-1}$ with a mean and standard deviation of $-255.02 \pm 57.7$ gC $m^{-2}yr^{-1}$, respectively (*Hollinger et al., 2016*), reflecting typical interannual ecosystem carbon dynamics. In contrast, the CAR681 model values for 2009 to 2013 were invariant, a trend not observed for NEE1 sites. The CAR681 subsequent year data are in error, on average by 50% less, compared to the Ho-1 NEE1 data, an exception for the CARB-CAR population, a trend resulting from selection of model parameters. Ho-1 NEE for the 2008–2013 period totals $\sim13{,}446$ $tCO_2e$ in contrast to 48,852 $tCO_2e$ based on the CARB-CAR protocol. Ho-1 NEE data for the period 1996 to 2002 and 1996 to 2008 (*Hollinger et al., 2004*; *Hollinger et al., 2013*) have not been compared to CAR681 data. CAR681 results ($-994.0 \pm 2{,}126.1$ gC $m^{-2}yr^{-1}$) do not reflect Ho-1 directly measured mean or interannual values ($-255.02 \pm 57.7$ gC $m^{-2}yr^{-1}$) and cannot be ignored as criteria for invalidation. However, if CARB-CAR data is interpreted as approximating GPP (e.g., not NEE), and $\sim82\%$ of assimilated carbon is lost as ecosystem respiration (*Hollinger et al., 2013*; *Hollinger et al., 2016*; NEE1,2), an approximate correction to the CARB-CAR data reduces the mean Howland NEE1 and CARB-CAR population values to $\sim-179$ and $\sim-170$ gC $m^{-2}yr^{-1}$, respectively, similar to the NEE1 (e.g., $-198 \pm 261.6$ gC $m^{-2}yr^{-1}$) and NEE2 (e.g., $-156 \pm 284$ gC $m^{-2}yr^{-1}$) population means. The high variance for interannual CARB-CAR results (e.g., $\pm 1504.8$ gC $m^{-2}yr^{-1}$) cannot be adjusted based on the correction, nor is the correction proposed as a universal revision to individual CARB-CAR projects. However, the adjustment generally supports the results presented here recognizing that $R_{eco}$ is absent from CARB-CAR carbon accounting (Supplemental Information 8). This comparison emphasizes the difficulty of determining soil carbon efflux and $R_{eco}$ terms with CARB-CAR protocol tools; the NEE method integrates GPP and $R_{eco}$ fluxes providing NEE (e.g., NEE1,2, Supplemental Information 9).

The financial and carbon market consequences for the CAR681 offset errors are substantial. Assuming a 2015 average price of $9.70 USD per offset credit ($tCO_2e$) (*Hammrick & Gallant, 2017*), sale of CAR681 offsets equal $\sim\$473{,}864$ compared to $130,431 for Ho-1 NEE based offsets, a difference of $343,433. Overpayment of $\sim4\times$ for Howland Forest offsets establishes carbon asset value loss and error in subsequent
financial transactions including a debit to the AB32 compliance carbon ledger. The initial vintage year overcrediting results from selectively establishing the Common Practice above-ground standing live carbon stock (e.g., 46 $tCO_2e$ $acre^{-1}$; 31 tC $ha^{-1}$) to ~4× below the independently determined value for the site (119–150 tC $ha^{-1}$; *Hollinger et al., 2013*). The CARB-CAR estimated baseline scenario maximizes credits generated in the initial year that cannot reflect known rates of $CO_2$ removal from the atmosphere (e.g., 5,332.3 gC $m^{-2}yr^{-1}$, ~26× in excess of the reported population annual mean NEE1 data of e.g., $-207.99$ gC $m^{-2}$ $yr^{-2}$, and ~34× the population annual mean NEE2 data of $-156$ gC $m^{-2}$ $yr^{-2}$). Lowering the Common Practice value relative to actual standing carbon stock for Howland Forest, incorrectly attributes historic carbon (e.g., unknown years of cumulative forest growth prior to the project) to a single vintage initial year of carbon sequestration. In effect, this results in higher initial year crediting, consistent with independent error analyses of similar CARB-CAR projects (*Dunlop, Winner & Smith, 2019*; *Haya, 2019*; *Haya et al., 2016*).

In contrast, Ho-1 NEE increased by ~6 gC $m^{-2}yr^{-1}$ over the last 19 years, representing ~50% overall increase of forest carbon sequestration; each year represents an annual baseline relative to net negative or net positive emissions to the atmosphere (*Hollinger et al., 2013*; *Richardson et al., 2019*). There is no initial-year front loading of historic carbon. The time series trend for Ho-1 would not be detected based on the CARB-CAR invariant annual data (excluding first year data), emphasizing the importance of trend detection sensitivity for carbon sequestering ecosystems. Ho-1 provides extensive data to test CARB-CAR protocols including process-based model development (*Sihi et al., 2018*), independent direct measurement of soil $CO_2$ efflux and ecosystem respiration (*Giasson et al., 2013*), contemporaneous NEE for $CO_2$, $CH_4$ and $N_2O$ (*Richardson et al., 2019*), response to shelterwood harvest (*Scott et al, 2004*) and diverse ecological data (*Giasson et al., 2013*). In summary, incomplete carbon accounting, model-based overcrediting carbon offsets by >5% relative to actual values, and demonstration that the CARB-CAR results do not agree with actual forest carbon sequestration, invalidate the CARB-CAR protocol for this project. Claimed reductions in net $CO_2$ emissions, carbon offset creation and carbon market transactions cannot be validated with the available data for CAR681 supporting a material misstatement calculation as defined in the AB32 rules (*California Air Resources Board, 2013b*) (endnote 2).

## CARB-CAR project review

Table 1 summarizes the CARB-CAR project sites ($n = 63$) and attributes considered in this study. Links to serial numbers for specific year vintage offsets issued and to summary pages of the CAR online documentation are provided. Features of the CARB-CAR carbon accounting process are identified and documented: (1) exclusion of the soil carbon pool and soil and ecosystem $CO_2$ emissions as respiration for the 63 CARB-CAR projects (Methods, CARB Protocols) cited in this study, (2) projects that record and assign serial numbers to offsets for single vintage year carbon sequestration but are based on records of partial or multiple years of carbon sequestration (63 instances), (3) arbitrary model operations initiated and executed as forward and/or backward runs relative to the project

[5]Carbon markets ideally conserve natural resources (*Osborne & Shapiro-Garza, 2018*; *Warren-Thomas et al., 2018*), limit surface warming to <1.5 °C relative to the pre-industrial period (*IPCC, 2018*; *Mengis et al., 2018*; *Rogelj et al., 2018*), and are commercially viable (*Sandor, Walsh & Marques, 2013*; *Rogelj et al., 2016*; *Bryant, 2018*). However, uncertainty in carbon product asset value (*Dhavale & Sarkis, 2018*; *Foss, 2018*; *Dunlop, Winner & Smith, 2019*) and market function can negatively affect carbon markets and their efficacy to manage climate change (*Helleiner & Thistlethwaite, 2013*; *Mehling, Metcalf & Stavins, 2018*; *Morrissey, 2018*; *Zhang, Randhir & Zhang, 2018*). For example, global carbon compliance markets have declined from ~$95B €in 2011 (*Thomson Reuters, 2016*) to $41B € in 2017 (*Thomson Reuters, 2018*), a decrease of ~57%, attributed to the absence of a price for carbon (*Johannsdottir & McInerney, 2016*), oversupply of offsets (*Ervine, 2018*; *Schatzki, Stavins & Hall, 2018*), ambiguity of disparate trading platforms (*Green, 2017*), and as we argue here for forest carbon, absence of direct and verifiable measurement of $CO_2$ and related carbon storage products (*Nisbet & Wiess, 2010*; *National Research Council, 2010*; *Peters et al., 2017*; *Palmer et al., 2018*). This uncertainty is reflected in global carbon markets and platforms. Macroeconomic trends for voluntary carbon trading markets are reflected in REDD+ verification programs. For example, 2016 prices for forest carbon were the lowest for REDD+ projects, averaging $4.60 $tCO_2$e on the largest volume of all project types (*Hammrick & Gallant, 2017*). VCS protocol projects were characterized consistently by the lowest pricing of $4.10 $tCO_2$e on the largest volume of all standards employed for forest projects (*Hammrick & Gallant, 2017*). Transaction volume (millions tons $CO_2$e) for forest carbon offsets fell ~40% from 2014 to 2016 (*Hammrick & Gallant, 2017*). We suggest that the low prices for REDD+ and VCS are, in part, related to the uncertainty and risk of invalidation for net carbon sequestration (*Foss, 2018*), as demonstrated for Howland Forest. In contrast, CARB-CAR pricing of ~$9.70USD $tCO_2$e for 2015 compliance offsets (*Hammrick & Gallant, 2017*), emphasize the asymmetry in carbon pricing; similar uncertainties apply to REDD+ (e.g., voluntary) and to CARB-CAR (e.g., compliance) forest carbon offsets.

start date (33 instances), and (4) excess initial year project carbon sequestration values (31 instances). Considerable discretion by CARB-CAR stakeholders appear to explain protocol inconsistencies; documentation for revisions is not provided. Protocol discretion, including increasing acceptance of default values for protocol components have been documented for related protocols (*Kollmuss & Fussler, 2015*) consistent with CAR -CAR protocol observations reported here.

## Survey of CARB-CAR linked protocols

Similar expressions are employed in the ACR, CDM and VCS protocols (Supplemental Information 8) extending the CARB-CAR carbon accounting uncertainty to large-scale forest carbon projects such as the UN-REDD and REDD+ (United Nations Framework Convention on Climate Change (*UNFCCC, 2019*), herein referred to as REDD+. REDD+ approved projects rely on the VCS (Supplemental Information 8). Methodologies developed under the United Nations CDM accepts projects and programs registered and approved by the VCS such as method VM0007 REDD+ Methodology Framework (REDD-MF), v1.5 (*VERRA, 2015*). Technical reports for REDD VCS applications categorically exclude forest soil carbon and ecosystem respiration (e.g., AC-6) from carbon accounting (*UNFCCC, 2017*) (Table 1). The implementation of REDD in Ghana, Africa, for example, is subject to impacts of potential invalidation for projects (*Asante et al., 2017*; *Kagombe et al., 2018*; *McFarland, 2018*) including World Bank sponsored bond programs similar to that operating in Kenya, such as the Kasigau Corridor project (*McFarland, 2018*), also based on VCS protocols. Results for REDD+ programs appear to be uncertain, in part, due to carbon offset monitoring and economics (*Foss, 2018*). Incomplete carbon accounting implies that results for REDD+ net forest carbon sequestration may not be verifiable or capable of identifying net annual ecosystem carbon change in response to reduced deforestation, climate and anthropogenic forcing, and may not be well suited for carbon pricing and trading of carbon financial instruments. Despite carbon accounting limitations, compliance carbon trading platforms and pricing initiatives are rapidly expanding (e.g., 45 national, 25 subnational jurisdictions; *World Bank and Ecofys, 2018*) emphasizing the importance of improved, shared methodology for forest carbon sequestration product offerings. Although it is not clear how REDD+ will be integrated within the Paris Agreement (e.g., Article 6) (*Schneider & La Hoz Theuer, 2018*) or into existing compliance markets (*Hein et al., 2018*), improved quantification of forest carbon sequestration would link REDD+ entities and mechanisms together in a harmonized universal science-based transactional framework. Macroeconomic trends consistent with offset invalidation risk for voluntary and carbon trading markets are provided in endnote.[5]

## CARB-CAR CA site data

The CARB-CAR project sites within California, representing 31 of the 63 projects (Table 1), represent a mean and standard deviation of −1,163.2 ± 1,889.8, respectively, more negative and variable than values for the CARB-CAR population data set (63 projects) of −948.8 ± 1,504.8 gC $m^{-2}yr^{-1}$. The CA sites establish the anomalous nature for this group relative to NEE1 (−198.0 ± 261.6 gC $m^{-2}yr^{-1}$) (Supplemental Information 9) and NEE2

($-156.0 \pm 284.0$ gC m$^{-2}$yr$^{-1}$) with up to ~7× the mean and ~12× the standard deviation (e.g., interannual variance) relative to NEE2 population values. The CARB-CAR sites are located generally within the western mountain biome and the marine coastal zones. The CARB-CAR biomes extend from southern to northern California to Oregon and Washington (Supplemental Informations 1, 9, filled gray rectangle symbol), with similar ecological patterns to support characteristic forest species. The CARB-CAR CA sites report invariant values (i.e., interannual variance of 0.0) for CAR590 (5 of 5 years), CAR694 (5 of 7 years), and CAR935 (12 of 14 years), that do not reflect natural forest ecosystems. Direct overlap of CARB-CAR and NEE methods is not required to establish the anomalous features of forest growth implied by the results for CARB-CAR CA sites relative to known population values. An accurate measurement of net carbon sequestration for any forest location is expected to comply with the GPP vs R$_{eco}$ relationship for NEE1 (Supplemental Information 9, filled black and gray rectangle symbols for west and east coast forest values as described) and NEE2 (e.g., $-156.0 \pm 284.0$ gC m$^{-2}$yr$^{-1}$). The CARB-CAR CA project sites according to the statistical analysis of NEE1 sites (e.g., Figs. 1–3, Table 2) and comparison with NEE1,2 data, indicate that the CARB-CAR CA sites do not represent realistic forest carbon sequestration and are arguably invalid.

## DISCUSSION

The offset errors, discretionary protocol inconsistencies, incomplete carbon accounting and statistical differences between the Howland Forest (CAR681) project data and directly measured contemporaneous Ho-1 NEE data pose an insuperable factual challenge to the scientific, regulatory and financial validity of the CARB-CAR protocol process and CARB compliance offsets. The consequences of erroneous financial transactions (i.e., ~4× overpayment), overcrediting to compliance carbon markets, early project termination, and loss of opportunity to manage net removal of atmospheric CO$_2$ do not satisfy the stated AB32 policy goals to "quantify net carbon sequestration as real, additional, permanent, verifiable, and enforceable" (*California Air Resources Board, 2011*; *California Air Resources Board, 2014*; *California Air Resources Board, 2015b*; *Nunez, 2016*). Net GHG removal, while stated as the objective of the CARB-CAR protocol, falls short of including established carbon accounting protocol terms for soil and ecosystem carbon efflux to achieve the required result (e.g., RF-6, IFM-6, AC-6, Methods 2.1, Table 1). The CAR681 project offsets exceed the allowable 5% threshold for overcrediting error (i.e., section 95985 of the AB32 regulation) rending buffer pools of limited value in compliance enforcement. The CAR681 data (i.e., anomalous initial year value, invariant subsequent years) has no counterpart in nature and is arguably invalid on this basis. The CARB-CAR Howland Forest project did not result in a long-term comparison between the protocols; 2014 was the last vintage year reported. In contrast, NEE data collection for Howland Forest has continued, comprising one of the longest NEE time series on record (1996-present) (*AmeriFlux, 2019*) documenting interannual changes in forest carbon storage balance between GPP and R$_{eco}$ (*Baldocchi & Penuelas, 2019*). The well-established anomalous nature of CARB-CAR CA projects ($-1,163.2 \pm 1,880$, $n = 31$), do not require site-specific overlapping NEE data

for comparison with NEE1,2 population data to conclude that the CA site data do not reflect known ranges for NEE and are arguably invalid, consistent with systemic errors noted for Howland Forest (e.g., 31 cases of anomalous initial year values, Table 1). Cases for invalidation based on problems of additionality and leakage (*Dunlop, Winner & Smith, 2019*; *Haya, 2019*) as well as questions raised by extension of CARB-CAR protocols to REDD+ tropical forests ('Survey of CARB-CAR Linked Protocols', endnote 5) (*California Air Resources Board, 2018b*), underscore the need and importance of resolving CARB-CAR protocol uncertainties.

Exclusion of the soil carbon and ecosystem respiration fluxes impose additional invalidation uncertainty for CARB-CAR protocols by the requirement of a 100-year invariant project baseline to ensure forest carbon storage permanence (*California Air Resources Board, 2011*; *California Air Resources Board, 2014*; *California Air Resources Board, 2015b*; *Climate Action Reserve, 2018b*). Soil carbon comprises up to three times the magnitude of above ground carbon composition, contributes up to ∼82% of ecosystem carbon exchange (*Baldocchi & Penuelas, 2019*; *Barba et al., 2018*; *Giasson et al., 2013*; *Hollinger et al., 2013*) and cannot be excluded from a complete, scientifically valid, carbon sequestration value for a forest project (*Comeau et al., 2018*; *DiRocco et al., 2014*; *Li et al., 2018a*; *Li et al., 2018b*). Soil warming and related soil $CO_2$ efflux predictions, including feedbacks to the biosphere (*Davidson & Janssens, 2006*), vary over the coming decades (*Bond-Lamberty et al., 2018*; *Hicks Pries et al., 2017*; *Melillo et al., 2011*; *Wang et al., 2014*; *Yang et al., 2013*) but they typically deny the assumption that the soil carbon pool and resulting fluxes will remain invariant over the 100-year required project interval (*Bond-Lamberty et al., 2018*; *Li et al., 2018a*; *Li et al., 2018b*). The global soil-to-atmosphere (e.g., total soil respiration) $CO_2$ flux, driven by climate change, is increasing across diverse contemporaneous ecosystems (*Bond-Lamberty et al., 2018*), a trend supported by a series of NEE observation platforms, inclusive of 19 of the 59 NEE1 sites represented in this study (*Baldocchi, 2008*; *Baldocchi, Chu & Reichstein, 2018*; *Bond-Lamberty & Thomson, 2010*). The CARB-CAR estimated invariant baselines, and the invariant interannual records (Table 1) have no counterpart in nature and should be replaced by direct measurement. The omission of soil-based $CO_2$ efflux for CARB-CAR and related protocols preclude differentiation of net-negative to net-positive $CO_2$ forest emissions, a critical test for forest carbon protocols, and a criterion for invalidation.

The CARB-CAR and NEE1 comparison establish systemic uncertainty and corrective action that apply broadly to the CARB-CAR and linked protocols (e.g., ACR, CDM, VCS) (Supplemental Information 8), including: (1) absence of direct, high frequency molecular $CO_2$ measurement to determine annual NEE, a requirement for project validation compliance testing and enforcement, corrected by inclusion of $CO_2$ flux measurements, (2) incomplete carbon accounting by exclusion of GPP and $R_{eco}$, corrected by requiring inclusion of soil carbon and ecosystem respiration terms, (3) discretionary adjustment of model growth forecasts (e.g., selection of FVS values) to achieve specific project goals including baseline levels, corrected by regulation of protocol rules, (4) creation of excess initial-year project vintage carbon sequestration values exceeding the known range of NEE values, violating the 5% invalidation threshold, corrected by using

actual NEE measurement for year one in the same manner as for all years, and (5) creation of unverified forest carbon financial products for carbon market transactions, corrected by employing third party verifiers with independent $CO_2$ measurements.

The protocol inconsistencies and invalidation criteria noted above (e.g., Howland Forest) support results of statistical analyses for CARB-CAR and NEE1 population comparisons. Mean values, Figs. 1A and 1B, and discrete interannual segments including exclusion of initial year data (e.g., Figs. 2 and 3, low $p$-values Table 2) were compared with the same result: the CARB-CAR protocol cannot be relied upon to represent valid net annual forest carbon dynamics and will likely exceed the 5% compliance threshold. Results for CARB-CAR net carbon sequestration cannot be argued as exceptions to known variance for forest populations (i.e., mean and standard deviation), but rather stand clearly outside of the known boundaries (Supplemental Information 9). We argue that the diverse analyses reported, considered together, provide robust observational evidence that the CARB-CAR and related protocols do not reliably reflect known values for forest carbon sequestration and are invalid.

The problems with estimation-based protocols can be addressed with existing technology and scientific methods. In contrast to estimation-based protocols, NEE for $CO_2$ determined by eddy covariance integrates vertical carbon fluxes between forest and soil ecosystems and atmosphere (e.g., assimilation and respiration) (*Burba, 2013*), resulting in the key outcome of net forest carbon sequestration (e.g., NEE as gC m$^{-2}$yr$^{-1}$). The eddy covariance method employed at the Howland Forest and NEE1,2 sites has been applied worldwide as standalone field research installations (*Burba, 2013*; *Curtis & Gough, 2018*; *Aubinet, Vesala & Papale, 2012*; *Lee, 2017*) in combination with remote sensing (*Hopkinson et al., 2016*; *Liu et al., 2016*) and as research networks (*FLUXNET, 2019*; *Novick et al., 2018*; *Ocheltree et al., 2007*; *Papale & Valentini, 2003*) for $CO_2$, $CH_4$ and $N_2O$, capability that cannot be implemented by CARB-CAR and related protocols. Eddy covariance networks are not typically interconnected in real-time or applied across large project areas for creating commercial forest carbon financial products. Commercial engineering development of low-cost eddy covariance networks, to attain state-wide spatial coverage across the US or for specific regions, for example, coupled with innovative features, including unmanned aerial vehicles (*Berman et al., 2012*; *Metzger et al., 2012*), shared data networks (*Dai et al., 2018*) and automated reporting is achievable offering a modernized alternative to estimation protocols. Advancements in blockchain accounting platforms (*Düdder & Ross, 2017*), artificial intelligence (*Reis et al., 2018*) and the internet of things (*Subashini et al., 2018*) can be readily integrated within eddy covariance networks but for the reasons we discuss here cannot be successful without direct measurement of $CO_2$. Eddy covariance as an instrumental method has characteristic limitations and uncertainties (*Nicolini et al., 2018*) and faces engineering challenges for large-scale deployment as well as upscaling eddy covariance methods (*Barba et al., 2018*; *Kumar et al., 2016*; *Ran et al., 2016*; *Warner et al., 2019*; *Wutzler et al., 2018*). NEE platforms and networks are established in numerous countries including those with REDD+ platforms and commitments to the Paris Agreement providing a foundation for expanded NEE observations to support global climate change policies.

Given the feasibility of establishing eddy covariance networks, expanding carbon market exchanges (*World Bank and Ecofys, 2018*) and the abundance of deforested landscapes, we suggest that standard methods and protocols be adopted for forest carbon and GHG financial products that: (1) are based on direct measurement of molecular $CO_2$ forest flux, (2) employ shared gas standards for $CO_2$ analyzer calibration, performance testing (e.g., World Meteorological Organization; *Brailsford, 2012*) and global reference frameworks (*Andrews et al., 2019*), (3) employ standardized protocols, model parametrizations and criteria such as that established by the Integrated Carbon Observation System (*Kutsch et al., 2018*; *Vitale & Papale, 2017*; *Wutzler et al., 2018*) (ICOS) and the Global Atmosphere Watch (*Schultz et al., 2015*), and (4) establish universal measurement-based criteria for the transformation of NEE to forest carbon offset products suitable for carbon financial transactions establishing equivalence between voluntary and compliance forest carbon products.

We acknowledge the limitations of this study related to the small sample sizes for annual intervals presented and limited overlapping project sites. However, in the absence of CARB-CAR validation and apparent protocol discrepancies reported here, NEE offers the best available data for independent verification. The results presented here form the basis for ongoing comparison between CARB-CAR and NEE1,2 results. CARB-CAR CA sites, likely to increase based on AB398 legislation (*Schatzki, Stavins & Hall, 2018*), we suggest, should incorporate current and evolving NEE methods, terminology and knowledge of forest carbon sequestration in collaboration with NEE research efforts.

## CONCLUSIONS

In conclusion, our results, unless proven otherwise, call into question the scientific, regulatory and financial validity of estimation-based protocols for net forest carbon. Diverse criteria demonstrate that CARB-CAR protocols are unreliable and unlikely to represent actual carbon sequestration and are invalid. The importance and urgency of validating forest carbon sequestration cannot be overestimated as forests provide additional ecosystem services of soil carbon sequestration and water conservation (*Masciandaro et al., 2018*), biodiversity safeguards (*Keesstra et al., 2018*), and the coupling of avoided forest carbon emissions with Indigenous Peoples habitation (*Mesoamerican Alliance of Peoples and Forests, 2015*; *Ramos-Castillo, Castellanos & Galloway McLean, 2017*). Measurement and standardized methods are hallmarks of the Montreal Protocol with 197 national signatories to monitor and enforce the reduction in emission of chlorofluorocarbons demonstrating the success of collective action within a common standardized analytical framework (*Bielewski & Śliwka, 2014*; *Hurst, Bakwin & Elkins, 1998*; *Newman, 2018*). Nothing less is required to advance forest carbon management and carbon market value. The demise of the Chicago Climate Exchange (*Sabbaghi & Sabbaghi, 2017*), coincident with near zero-dollar value for forest carbon (*Peters-Stanley et al., 2012*), and false claims of automotive emission avoidance offers a lesson learned that without actual measurement and accountability, institutional GHG emission reduction frameworks are vulnerable to economic and policy failure (*Van Renssen, 2018*) and fraud (*Li et al., 2018a*; *Li et al., 2018b*).

Project specific climate finance and monetization mechanisms are a key but unspecified component of the Paris Agreement that if combined with direct measurement of forest $CO_2$ will benefit societies and economies in the coming decades and prove crucial to correcting the imbalance between nature and anthropogenic activity and resulting climate change.

## ACKNOWLEDGEMENTS

The references cited in this study referring to eddy covariance data were acquired and shared by the FLUXNET community, including potentially these networks: AmeriFlux, AfriFlux, AsiaFlux, CarboAfrica, CarboEurope-IP, CarboItaly, CarboMont, ChinaFlux, FLUXNET Canada, Green- Grass, ICOS, KoFlux, LBA, NECC, TERN OzFlux, TCOS-Siberia, and USCCC. Detailed data (e.g., annual records) cannot be shared publicly because of Fluxnet2015 (https:// fluxnet.fluxdata.org/data/data-policy/) and Lathuile (https://fluxnet.fluxdata.org/data/la-thuile-dataset/) data policies. Annual values used in this study were acquired from individual references cited in NEE1 (*Baldocchi, Chu & Reichstein, 2018*).

### Funding

The authors received no funding for this work.

### Competing Interests

Bruno D.V. Marino is an Academic Editor for PeerJ. Bruno D.V. Marino and Aaron Doucett are unpaid associates of Planetary Emissions Management Inc.

### Author Contributions

- Bruno D.V. Marino conceived and designed the experiments, analyzed the data, prepared figures and/or tables, authored or reviewed drafts of the paper, approved the final draft.
- Martina Mincheva analyzed the data, prepared figures and/or tables.
- Aaron Doucett prepared the interactive map.

### Data Availability

The raw data for NEE and CARB-CAR sites are available in the Supplemental Files.

### Supplemental Information

Supplemental information for this article can be found online at http://dx.doi.org/10.7717/peerj.7606#supplemental-information.

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
