# Peer review of "California air resources board forest carbon protocol invalidates offsets"

_PeerJ, doi:10.7717/peerj.7606_

## Round 0.1 · original submission · Major Revisions

We received one "reject" (R2) and two "major revision" recommendations for your manuscript. All three reviewers make excellent comments and suggestions which I recommend you follow closely if you choose to send a revised version in. Two general critiques I also have is the mismatch in CARB and FluxNet sites and the need for a shorter, more direct text.

Reviewer 1 ·

Basic reporting

Line 83, "Deforestation continues at a rate..." for this sentence, read this paper: "High-Resolution Global Maps of 21st-Century Forest Cover Change" by Hansen et al 2013

Line 109, was this forest survey the FIA dataset? I thought trees were surveyed more often than six years.

Line 112, is there any studies evaluating the limitation/uncertainty of CARB? Since this is the review paper, it would be good to mention other existing papers addressing the similar questions and what values of this paper adding to it. Or is this paper the only study reporting the uncertainty?

Line 126, yes, NEE is kind the same as NEP, but NEP = -NEE

The reference format (both in-text citation and reference list) is not correct, please check "Instructions to authors"

Figure 3, if there is no negative values, why extending y axis to below zero?

Experimental design

Following the "Instructions to authors" on Literature Review format, The section of "4.0 Survey and Analysis Methods" should be moved up to the second portion.

And there is another section- "Statistical Methods"-appears after "Acknowledgement" in the end. Should this be moved up to the "Survey and Analysis Methods" or be inserted in the Appendix?

Line 667, authors should change the subheading "data sources"to another because below authors have "CARB-CAR Data Sources" and "NEE Data Sources", so this is a summary of all the data sources?

Line 698-699, it might not be necessary to list all the 30 numbers out.

Line 741, were listing those nine columns here necessary? were those all useful for readers to understand the paper?

Line 803, extra indent, not necessary.

Line 812, there are lots of other papers reporting the DBH and biometric equation uncertainty, not just one paper from tropical forest cited here. If authors want to summarize the magnitude of uncertainty, it is better to add more references. Some others are:
Paré, D., G.Z. Gertner, P.Y. Bernier, and R.D. Yanai. 2016. Quantifying uncertainty in forest measurements and models: Approaches and applications.Canadian Journal of Forest Research 46(3): v. 10.1139/cjfr-2016- 0029
Holdaway, Robert J., Stephen J. McNeill, Norman W. H. Mason, and Fiona E. Carswell. 2014. Propagating Uncertainty in Plot-based Estimates of Forest Carbon Stock and Carbon Stock Change. Ecosystems in press. DOI: 10.1007/s10021-014-9749-5
Or if you go to this Ruth Yanai's website "http://quantifyinguncertainty.org/", click "Bibliography-Biomass Uncertainty", there is more.

Line 830-831, this sentence is confusing, "59 sites covering 540 sites..?" Or do authors mean 59 towers?

Validity of the findings

From line 160-192, the first two paragraphs in the "2.Results" are all Methods. I did not see any sentences describing the results here. Just about what methods/steps authors have conducted and in what tables/graphs.

Line 193-197, those sentences are describing the box plot legend, can be inserted in the figure caption, but not here. And same thing for Line 236-239

Line 209, delete extra space

Line 273, why p values being differed among years? were they discussed in the Discussion section?

Line 373, this is an important point. It might be better for authors to add some references reporting estimated soil carbon pool and see how it looks if being added to CARB-CAR.

Line 381, it is also important to discuss some limitations on upscaling eddy fluxes tower data.

Line 429, should part of this section be moved up to the Result section? as a special case for comparing CARB-CAR with NEE1. Because the the limitations discussed above this paragraph-"Protocol Process" and "Soil Carbon Exclusion' are also applied here.

Line 475, any ideas about why the forest survey had larger AGB than CARB-CAR estimation?

Line 580, this is good point. Right now most of the eddy flux tower are run for academic research and supported by NEON. In California, most of the tower are in southern region (dry) but not in northern region (wet), causing a bias if upscaling the results to the whole state.

Additional comments

This paper reviews forest carbon protocols and offsets issued by CARB-CAR, and compared with measurements of annual net ecosystem exchange from eddy flux towers. It reports the large variation of estimated forest carbon within each protocol, and a 5 times differences in annual value comparing the two protocols. This paper adds a great value, connecting economy to academia. It also adds great thoughts on the current application of those protocols.

Reviewer 2 ·

Basic reporting

see attachment

Experimental design

see attachment

Validity of the findings

see attachment

Annotated reviews are not available for download in order to protect the identity of reviewers who chose to remain anonymous.

Reviewer 3 ·

Basic reporting

Review of ‘’California air resources board forest carbon protocol invalidates offsets”



Carbon markets remain largely a panacea for contending with climate change, though some nations and states (such as California) are entering into cap and trade agreements. Projects that are designed to defer deforestation or regain C through forest management underlie many of the schemes.

This manuscript looks at projects adopted under the California Air Resources Board. I do not confess to understand how these projects are designed or managed to receive C credits, though the authors here outline the forest measurements and simulation models. The authors of this manuscript are concerned that these projections may be flawed or unrealistic, and thus compare C sequestration rates from the CARB protocols to those measured more accurately through net ecosystem CO2 exchange with data from the FluxNet network.

The major problem with this comparison is that the CARB and FluxNet measurements are conducted at different sites, with the exception of Howland Forest, in the NE. Figure 1b, for example, is quite interesting, but we do not know if some unique management is being conducted at the CARB sites to facilitate the larger negative emissions, or if it is calculation error.

The authors point out that the CARB C credits at Howland show a unrealistically large rate the first year (2008, I think). I wish they would have clearly identified why that occurred. But, after that, the rates drop down to ~ FluxNet values. Thus, I was not clearly given illumination a comparion of this key site.


It seems the authors are motivated by the reasonable concern about the validity of C credits on any C market. That is a very fair concern. This manuscript doesn’t quite achieve this:

1. It is incredibly long, and dense. Shed a lot of the verbiage, and focus on the CARB vs FluxNet at Howland as an excellent comparison. Why does CARB give a huge credit at teh beginning? How do they compare over time after that?
2. Then, if the authors wish, compare CARB project to FluxNet, but compare projects that have similar site history to FluxNet, so that sites recoverying from disturbance or from forest management enhancements are not skewing rates.
3. The soil C issue seems out of place here. I can understand why CARB just by passes it. It IS embedded to some degree in NEE, which makes the two metrics somewhat different.
4. I think Supplement 2 overdoes the summation equations, but under-explains the CARB process.



To summarize, Fig. 3 underscores the legitimate concern of this paper, but more importantly, the concept that C can be stored and traded in natural ecosystems. Are the estimates realistic? Does it have climatic significance? With clarification, considerable editorial streamlining and reorganization, this manuscript might offer an important reality check for CARB estimates. However, the manuscript as it now stands is not ready for publication, and needs to be reduced from a dissertation to a digestible article-length analysis.

Experimental design

see above

Validity of the findings

see above

Additional comments

none

---

## Round 0.2 · Minor Revisions

Two of the three previous reviewers agreed to evaluate your re-submission. Reviewer 1 is happy with the manuscript as is. It seems clear to me that the issues raised by Reviewer 2 have more to do with (healthy) skepticism about the carbon market overall and not the technical or scientific merits of your work. In my view, your manuscript offers a valid contribution to the field and would be in acceptable form once the small problems with Fig S7 are resolved. This is something we can sort out internally without the need for more external reviews.

Reviewer 1 ·

Basic reporting

None

Experimental design

None

Validity of the findings

None

Additional comments

I really appreciated authors' hardworking on revising this manuscript. I am satisfied with those responses to my comments.

Reviewer 3 ·

Basic reporting

The manuscript is now clearly written (though it still could be shorter) for a non-specialist to see what the objectives are, and how the data inform the conclusions. I have to say, the entire premise and framework of the CARB program is remarkably shaky and (if this review is correct) untested.

Experimental design

The premise here is to use publically available NEE data for ecosystems as a constraint on the likely range of NEE under CARB C storage projects, to see if CARB estimates of sequestration are realistic. As I mentioned last time, the CARB sites might start off in such a degraded state they may gain more C than the FLUXNET sites. I can't sort this out, and if the CARB has such information (or other sceintists/foresters), then they can reply. This paper at least opens up the dialogue for scientific evaluation of these schemes.

The paper has the benefit of a direct CARB to FLUXNET comparison the east coast, which at least provides some assurance that the approach here is valid. This site appears to support the contention of the authors.

Validity of the findings

Again, its the history of the CARB sites prior to the start of a project, but even so, there are natural limits to NEE.

The whole arena of C markets and natural storage projects is, to me, a strange and hard to understand space. As a C researcher, I dont have the time to wade through all the regulatory or contractal paperwork, and figure out what is going on. I dont know if ecosystem exchange studies can be conducted for every C sequestration project, but I do think more can be done to remove the black magic aspects.

Additional comments

On Figure S7, both in text and in teh figure caption, there are different statements about what the symbols are, adn what they mean. I dont see any gray circles in the figure, and it is not clear what the gray and black boxes are.

---

## Round 0.3 · accepted · Accept

Thank you for the clarity and objectivity of your response to reviewers documents.